# Sequential involvements of the perirhinal cortex and hippocampus in the recall of item-location associative memory in macaques

Cen Yang[1,2], Yuji Naya[1,3,4] *

**1** School of Psychological and Cognitive Sciences, Peking University, Beijing, China, **2** Center for Life Sciences, Peking University, Beijing, China, **3** PKU-IDG/McGovern Institute for Brain Research, Peking University, Beijing, China, **4** Beijing Key Laboratory of Behavior and Mental Health, Peking University, Beijing, China

* yujin@pku.edu.cn

## Abstract

The standard consolidation theory suggests that the hippocampus (HPC) is critically involved in acquiring new memory, while storage and recall gradually become independent of it. Converging studies have shown separate involvements of the perirhinal cortex (PRC) and parahippocampal cortex (PHC) in item and spatial processes, whereas HPC relates the item to a spatial context. These 2 strands of literature raise the following question; which brain region is involved in the recall process of item-location associative memory? To solve this question, this study applied an item-location associative (ILA) paradigm in a single-unit study of nonhuman primates. We trained 2 macaques to associate 4 visual item pairs with 4 locations on a background map in an allocentric manner before the recording sessions. In each trial, 1 visual item and the map image at a tilt (−90° to 90°) were sequentially presented as the item-cue and the context-cue, respectively. The macaques chose the item-cue location relative to the context-cue by positioning their gaze. Neurons in the PRC, PHC, and HPC, but not area TE, exhibited item-cue responses which signaled retrieval of item-location associative memory. This retrieval signal first appeared in the PRC, followed by the HPC and PHC. We examined whether neural representations of the retrieved locations were related to the external space that the macaques viewed. A positive representation similarity was found in the HPC and PHC, but not in the PRC, thus suggesting a contribution of the HPC to relate the retrieved location from the PRC with a first-person perspective of the subjects and provide the self-referenced retrieved location to the PHC. These results imply distinct but complementary contributions of the PRC and HPC to recall of item-location associative memory that can be used across multiple spatial contexts.

## Introduction

Semantic memory, the memory of factual knowledge, is a subcategory of declarative memory [1,2]. Semantic memory is acquired through repetitive experiences that share common facts in

**Data Availability Statement:** The source data for all figures in this manuscript are available as supporting information in S1 Data. The original data of spike are available at the following osf.io

repository: https://osf.io/vaet6/?view_only=
2a3fb059695143178b323b6c6cbcb028.

**Funding:** This work is supported by the Ministry of
Science and Technology of the People's Republic
of China, STI2030-Major Projects https://www.
most.gov.cn/ (grant number 2021ZD0203600).
Received by YN. It is supported by the National
Natural Science Foundation of China http://www.
nsfc.gov.cn/ (grant number 31871139). Received
by YN. It is supported by the National Natural
Science Foundation of China http://www.nsfc.gov.
cn/ (grant number 32150710526). Received by YN.
The funders had no role in study design, data
collection and analysis, decision to publish, or
preparation of the manuscript.

**Competing interests:** The authors have declared
that no competing interests exist.

**Abbreviations:** HPC, hippocampus; IACUC,
Institutional Animal Care and Use Committee; ILA,
item-location associative; MTL, medial temporal
lobe; NIH, National Institutes of Health; PHC,
parahippocampal cortex; PRC, perirhinal cortex;
RSA, representational similarity analysis; SDF,
spike density function.

various spatiotemporal contexts. In each experience, this common fact would eventually be
stored as semantic memory separately from the unique spatiotemporal context. This notion
contrasts with another subcategory of declarative memory (i.e., episodic memory), which
allows us to reexperience a particular past event [1]. Therefore, semantic memory is essentially
formed in an allocentric manner, while episodic memory accompanies the unique spatiotem-
poral contexts to reconstruct a particular personal event in the "self-referenced" manner or the
"first-person perspective" [1,3–6]. However, when we recall semantic knowledge, particularly
that containing spatial components, we may bring it to mind from a first-person perspective.
Assume a scenario where someone asks where the Statue of Liberty is in the United States;
when a map is presented, even if at a tilt, the Statue of Liberty can be easily located on it (Fig
1A), which would suggest an allocentric representation of semantic memory. In the absence of
the map, you may get mental imagery of the retrieved location to represent it from a first-per-
son perspective (e.g., rightward) by assuming a particular spatial context (e.g., top for the
north).

Concerning the responsible brain regions for the 2 subcategories of declarative memory,
previous neuropsychological studies have indicated the critical involvement of the hippocam-
pus (HPC) in encoding both new episodic memory and semantic memory, whereas HPC
lesions impair recollection of episodic memory but spare retrieval of semantic memory [7–11].
In contrast, the anterior temporal lobe, including the perirhinal cortex (PRC), is involved in
conceptual processing, which depends on factual knowledge stored as semantic memory. The
functional dissociation between the PRC and the HPC is also suggested by a series of dual-pro-
cess models in an item recognition paradigm [12–14]; the PRC contributes to recognition by
providing item familiarity, whereas the HPC contributes to recognition by recollecting the spa-
tiotemporal context in which the item was presented. These models are consistent with ana-
tomical evidence suggesting separate information processing along 2 distinct pathways from
neocortical areas to the HPC via the PRC and the parahippocampal cortex (PHC) and their
relational processing in the HPC [15–18]. The PRC receives visual input from the ventral path-
way including area TE, while the PHC receives visual inputs from the dorsal pathway in addi-
tion to the posterior part of the ventral pathway. The distinct mnemonic functions of the PRC
and the HPC in these 2 lines of preceding literature ("semantic memory vs. episodic memory"
and "familiarity/item vs. recollection/relation") raise the question of which brain area is
involved in the retrieval of semantic memory, particularly when it contains spatial
components.

To address this question, we used the item-location association (ILA) paradigm in macaque
electrophysiology [19]. In this study, before recording sessions, we first repeatedly trained
monkeys to associate visual items with locations on a background map image (Figs 1B and S1
and S2). Because the map image was presented at a tilt with a random orientation, the mon-
keys were encouraged to store a common relationship between the items and corresponding
locations relative to the map image across the past trials rather than to store a unique spatio-
temporal context in each trial. Thus, the long-term association memory tested by the ILA para-
digm could be classified as "semantic-like" rather than "episodic-like," even though it is
controversial whether nonhuman primates have semantic memory and/or episodic memory
[20,21]. In each trial, one of the learned visual items was presented as an item-cue, and then
the randomly tilted map image was presented as a context-cue. The monkeys made a saccade
to the location assigned to the item-cue relative to the context-cue (Fig 1C). We recorded spike
firings of single neurons during the monkeys performing the ILA task. We examined neuronal
responses to the item-cues in the absence of the map image (i.e., before the context-cue presen-
tation), which might remind the monkeys of locations associated with the item-cues. To evalu-
ate mnemonic components carried by neural responses to the item-cue, we used 2 types of

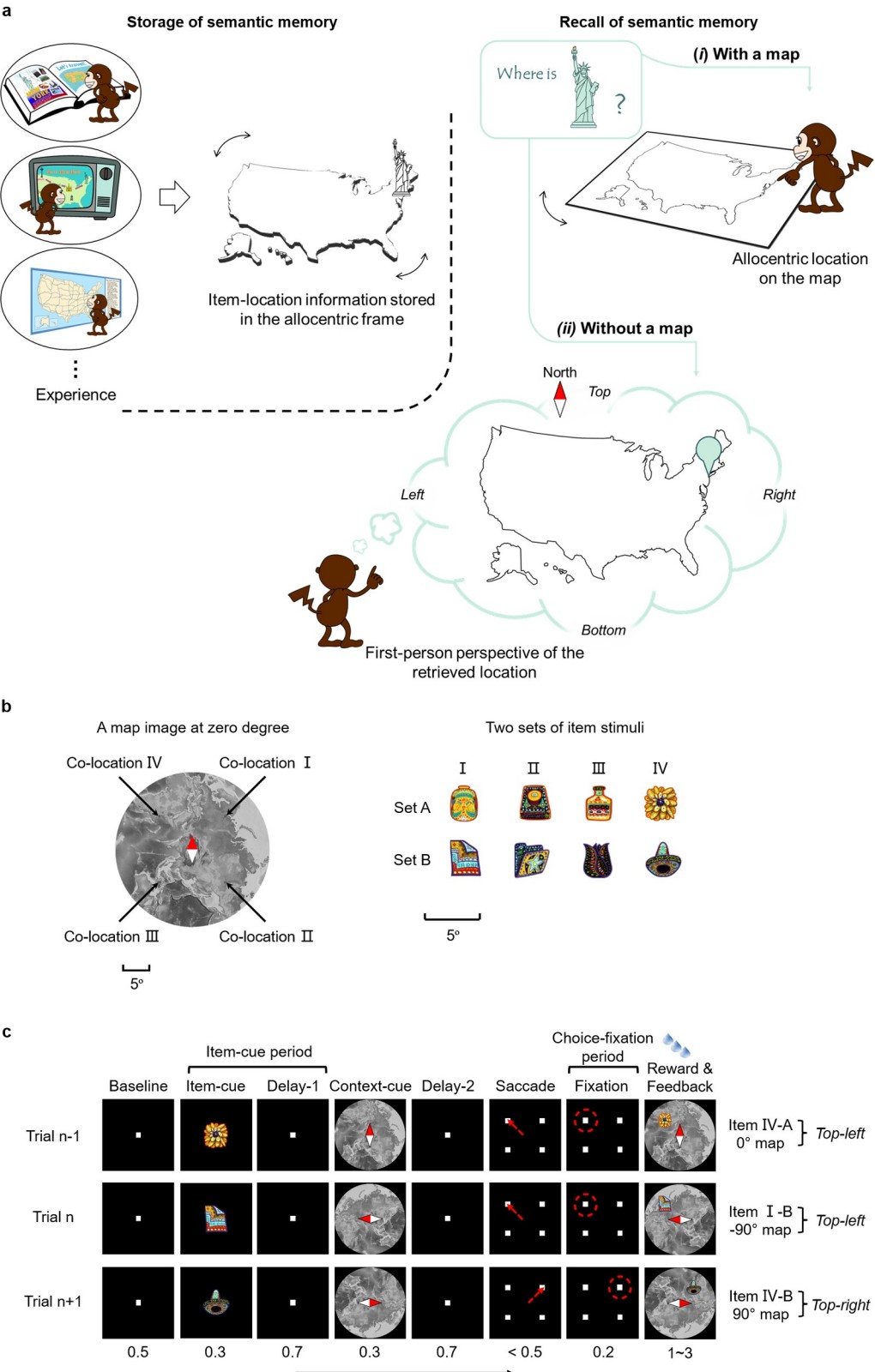

**Fig 1. Semantic memory including spatial components.** **(a)** (Left) Through repetitive experiences (e.g., reading a book in a study room, etc.), ILAs (e.g., Statue of Liberty ↔ northeast in the US) can be stored in the allocentric frame. The

stored memory, which does not accompany a unique spatiotemporal context from a single episode (e.g., watching TV in a living room), could be categorized into semantic memory. (Right) The recall process of the item-location associative memory may differ between the presence (*i*) and absence (*ii*) of a map on which the item's location can be shown. (*i*) When the map is present, the item can be located on it no matter how it is tilted (0° to 360°), suggesting the semantic recall in the allocentric frame. (*ii*) In the absence of the map, the item can be located in an assumed spatial context (e.g., top for the north), providing mental imagery of the retrieved location from the first-person perspective. The outline map image of the US was made based on an image from public domain, "Pixabay" (https://pixabay.com/illustrations/us-map-outline-us-map-america-1674031/). (**b**) ILA pattern. Two items, 1 from set A (e.g., I-A) and the other from set B (e.g., I-B), were assigned to each location (e.g., co-location I) on the map image. Scale bars for both item and map stimuli, with a 5° visual angle. The background map image was made based on an image (EMU 13) from public domain, "USGS" (https://www.usgs.gov/media/images/emu-13). (**c**) Schematic diagram of the ILA task. An item-cue and a context-cue were sequentially presented in each trial. The monkeys' gaze was fixated on the center until the end of the delay-2, then saccade to the target location (indicated by the red arrowhead and red dashed circle) according to the 2 cues. A successful trial was rewarded with juice paired with feedback showing the associated location of the item-cue on the context-cue. Relative sizes of the stimuli were magnified for display purposes. ILA, item-location associative.

measures. In the first measure, we tested the presence/absence of ILA effect by examining neurons' responses to pairs of the item-cues ("co-locating items") assigned to the same locations ("co-locations") on the map image. In the second measure, we evaluated the neural representations of the retrieved locations during the item-cue period by comparing with those of the target locations where the monkeys fixated during the choice-fixation period. The first measure revealed an earlier appearance of the item-location associative effect in the PRC before HPC. The second measure indicated that neural representations of retrieved item-cue locations were related with those of external space where the monkeys viewed, only in the HPC but not in the PRC. These results suggest that the PRC and the HPC are involved in remembering the semantic-like item-location associative memory in distinct manners. In addition to the PRC and HPC, we recorded signals from area TE (TE) of the ventral pathway and the PHC as control brain regions.

## Results

Two rhesus macaques were trained to perform the ILA task. In the ILA task, 4 visual item pairs ("co-locating items") were assigned to 4 different locations ("co-locations") on an image of a map [19]. The same 8 visual items and 1 map image were used during all the recording sessions (Fig 1B). In each trial, a randomly chosen visual item was presented as an item-cue. Then, the map image was presented at a tilt with a randomly chosen orientation (−90° to 90°, 0.1° step) as a context-cue (Fig 1C). By moving their gaze positions, the monkeys reported a target location (e.g., Top-left in the trial n, Fig 1C), which corresponded to a location assigned to the item-cue (e.g., co-location I) on the tilted map image of the context-cue (e.g., −90°). While the monkeys performed the ILA task, we recorded single-unit activity from a total of 1,175 neurons from the 4 brain areas (Fig 2A and S1 Table). During the recording session, the orientation of the context-cue was pseudorandomly chosen from among 5 orientations (−90°, −45°, 0°, 45°, and 90°). The task was correctly performed by both monkeys (chance level = 25%) at rates of 80.5% ± 8.4% (mean ± standard deviation; Monkey B, *n* = 435 recording sessions) and 95.1% ± 5.8% (Monkey C, *n* = 416 recording sessions). We used only correct trials for the following analyses. The HPC data was partly reported in a previous study [19].

### Retrieval of item-location associative memory

We analyzed the activities of single neurons during the item-cue period, including the presentation of the item-cue (0.3 s) and following delay-1 (0.7 s) periods, to examine their

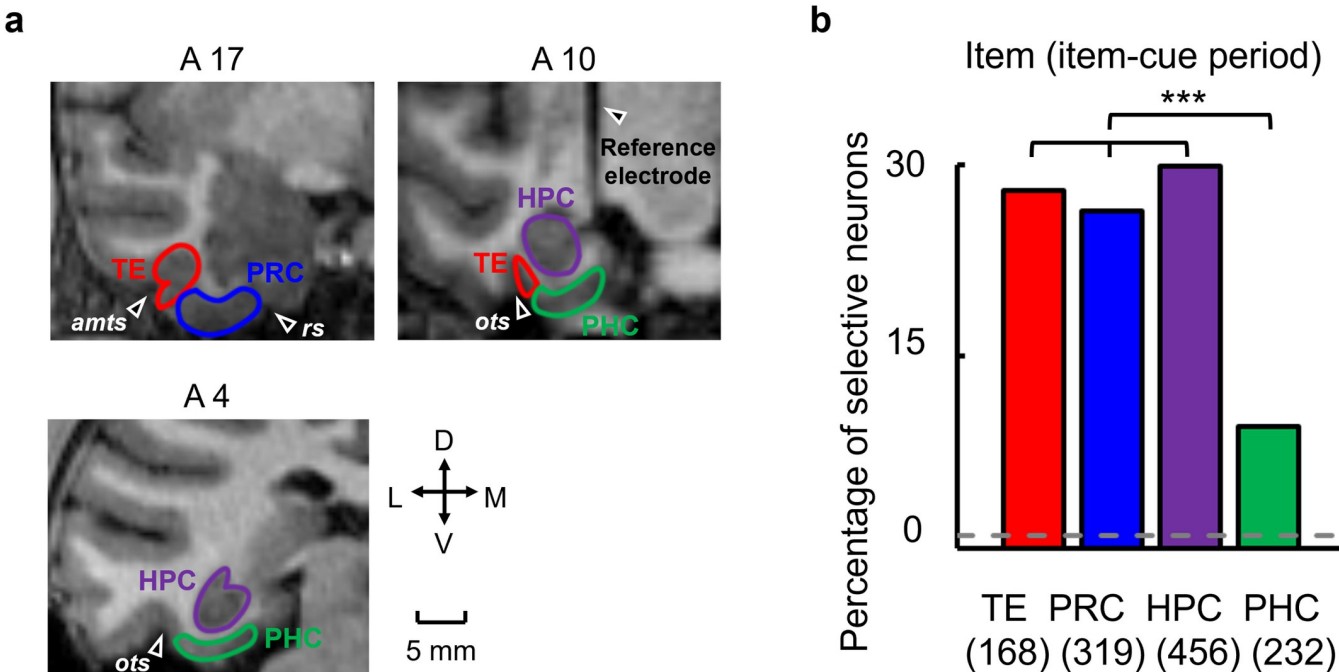

**Fig 2. Item-cue selective activities in the MTL and area TE (TE).** **(a)** Recording regions. MRI images corresponding to the coronal planes anterior 4, 10, and 17 mm from the interaural line of monkey C (right hemisphere). The recording regions are the PRC, HPC, and PHC of the MTL and TE. A reference electrode implanted in the center of the chamber was observed as a vertical line of shadow in the coronal plane at A10. amts, anterior middle temporal sulcus. rs, rhinal sulcus. ots, occipital temporal sulcus. D, dorsal. V, ventral. L, lateral. M, medial. **(b)** Percentage of item-cue selective neurons out of recorded neurons in each area. Parentheses, number of recorded neurons in each area. Dashed line, 1.0% chance level. The number of item-cue selective neurons was significantly larger than the chance level ($P < 0.0001$ for each area, one-tailed binomial test). Asterisk indicates results of a $\chi$-square test: PHC and TE, $P < 0.0001^{***}$, $\chi^2 = 23.3$, $d.f. = 1$; PHC and PRC, $P < 0.0001^{***}$, $\chi^2 = 24.5$, $d.f. = 1$; PHC and HPC, $P < 0.0001^{***}$, $\chi^2 = 36.0$, $d.f. = 1$. Source data are available in S1 Data. HPC, hippocampus; MTL, medial temporal lobe; PHC, parahippocampal cortex; PRC, perirhinal cortex.

involvement in item-location associative memory. We first selected neurons that exhibited differential item-cue responses among the 8 visual items (item-cue selective) at a threshold of $P < 0.01$ (one-way ANOVA). We found a large proportion of the item-cue selective neurons in the PRC (26.3%), TE (28.0%), as well as in the HPC (29.8%) (Fig 2B and S1 Table). In contrast, the proportion of the item-cue selective neurons was significantly smaller in the PHC (9.5%) than in the other 3 areas ($P < 0.001$ for each area, $\chi$-square test). This result was consistent with previous studies indicating preferential processing of item information in the ventral path-PRC stream [15–17,22].

Fig 3A shows an example of item-cue selective neurons from the PRC ($P = 0.0002$, $F(7,78) = 4.78$) (see S3 Fig for example neurons from TE, HPC, and PHC). This neuron exhibited the largest responses to 2 item-cues that were assigned to co-location II as well as the second-largest responses to 2 item-cues that were assigned to co-location III, indicating that similar response patterns were observed for the item-cues with the same co-locations ("co-locating items"). We evaluated the correlated responses to the 4 pairs of co-locating items by calculating the Pearson correlation coefficient between responses to item-cues "I-A"–"IV-A" and those to item-cues "I-B"–"IV-B" as the "co-location index." If a neuron showed a pattern of stimulus selectivity that was independent of the items' co-locations, the expected value of the neuron's co-location index was "zero." The co-location index of the example neuron shown in Fig 3 was significantly positive ($r = 0.98$, $P = 0.0024$, two-tailed permutation test), suggesting an effect of the item-location associative memory on its response pattern to the item-cues.

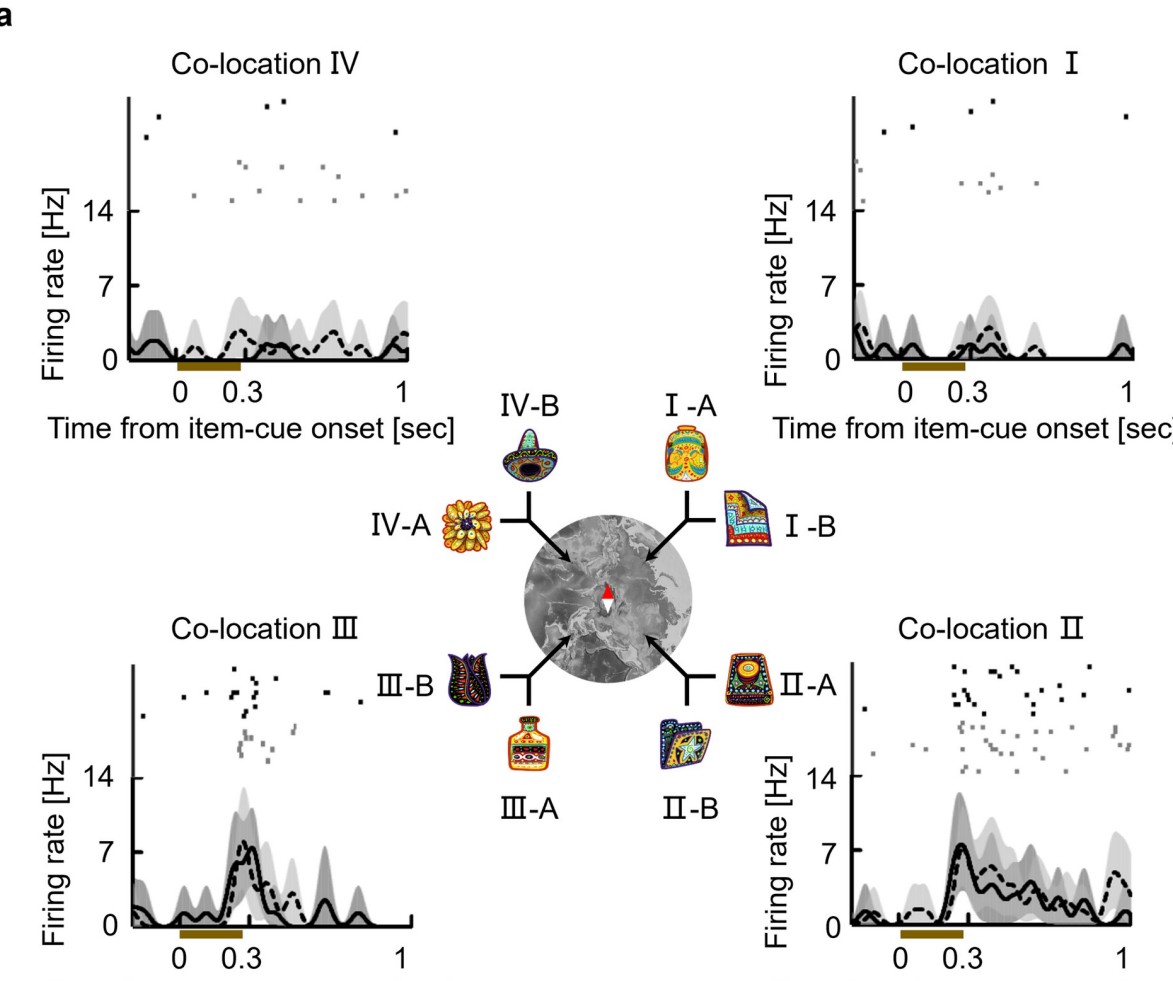

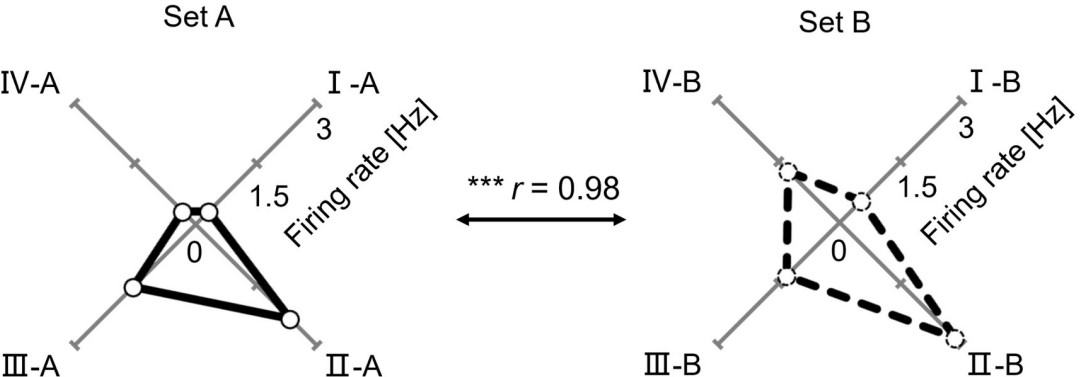

**Fig 3. A PRC neuron showing the co-location effect on item-cue selective activities. (a)** Solid lines and dashed lines indicate SDFs in trials with item-cues from the stimulus sets A and B, respectively. Dark and light gray shading, 90% confidence interval of 10,000 bootstraps (see **Materials and methods**) for the stimulus sets A and B, respectively. Black and gray dots, raster plots for the stimulus sets A and B, respectively. Brown bar, presentation of the item-cue. The background map image was made based on an image (EMU 13) from public domain, "USGS" (https://www.usgs.gov/media/images/emu-13). **(b)** Mean discharge rates of the neuron during the item-cue period for each item. *r*, correlation coefficient. Asterisk indicates the result of a two-tailed permutation test: *P* = 0.0024***. Source data are available in S1 Data. PRC, perirhinal cortex; SDF, spike density function.

We evaluated the association effect in each area by calculating the co-location index for each item-cue selective neuron. In addition to the HPC, which showed a high co-location index (median, $r = 0.89$, $P < 0.0001$, two-sided Wilcoxon signed-rank test) in our previous study [19], both the PRC and the PHC showed significantly positive co-location indices ($r = 0.63$, $P < 0.0001$ in PRC; $r = 0.63$, $P = 0.0009$ in PHC) (Fig 4A and 4B). In contrast to the striking association effect in the medial temporal lobe (MTL), TE did not show a significantly positive co-location index ($r = 0.16$, $P = 0.0754$), which is consistent with previous lesion studies indicating preferential involvement of TE in the perceptual processing of visual objects relative to the MTL (PRC) [23].

We next examined which area in the MTL first exhibited the association signal by comparing the time courses of the population-averaged co-location indices among the 3 MTL areas. To this end, we selected neurons with high co-location indices ($r > 0.8$), founding that the population-averaged co-location index increased earlier in the PRC ($n = 30$) than in the HPC ($n = 83$, $P = 0.0382$, two-tailed permutation test) (Fig 4C). The increase in the population-averaged co-location index in the HPC was followed by that in the PHC ($n = 8$), although the delay was not statistically significant ($P = 0.1048$, two-tailed permutation test). These results hold for neurons that were selected at different criteria ($r > 0.7$ or significant co-location effect, S4 Fig). The time course of the co-location index was also examined for each neuron with high co-location indices ($r > 0.8$) (see **Materials and methods** and S5 Fig). The half-peak times of the co-location indices for individual neurons were shorter in the PRC than in the HPC ($P = 0.0395$, KS = 0.29, Kolmogorov–Smirnov test) and PHC ($P = 0.0201$, KS = 0.57) (Fig 4D). Considering the dense fiber projections from the TE to the PRC [24,25], the perceptual information signaling the item-cue in TE may elicit a reinstatement of the item-location associative memory in the PRC, which is consistent with previous physiological studies examining other types of association memory paradigms [26–33].

## First-person perspective of the retrieved location

The item-cues elicited item-cue selective activity according to their co-locations, which were determined relative to the map image. However, it is yet to be addressed whether co-location-related activity signals the retrieved location in relation to a particular spatial context under real space. To answer this question, we compared the responses during the item-cue period with those during the choice-fixation period in which the monkeys fixated on a target location (Fig 1C). Before the comparison, we examined the representation of task-related information during this choice-fixation period by applying a three-way (item-cue, context-cue, target) ANOVA ($P < 0.01$) for each neuron in the MTL. A substantial number of neurons exhibited differential responses among the target locations (target-selective) in all the MTL areas (6.0% in PRC, 12.1% in HPC, and 10.8% in PHC) (S6A Fig, S1 Table), whereas only a negligible number (approximately 1.5%) of neurons showed item-cue or context-cue selective responses across the 3 subareas (S6A Fig and S1 Table). These results indicate that the MTL neurons represented the target location where the monkeys presently gazed during the choice-fixation period rather than the information from the preceding task events (i.e., item-cue and context-cue).

To compare the responses between the 2 task periods in each MTL area, we first examined whether neurons exhibiting item-cue selective responses during the item-cue period (i.e., item-cue selective neurons) also showed target-selective responses during the choice-fixation period. We found that the item-cue selective neurons had a significant tendency to show target-selective responses in the HPC (17.6%, $P = 0.0170$, $\chi^2 = 5.7$, $d.f. = 1$, $\chi$-square test) and PHC (27.3%, $P = 0.0087$, $\chi^2 = 6.88$, $d.f. = 1$), but not in the PRC (9.5%, $P = 0.1075$, $\chi^2 = 2.59$, $d.$

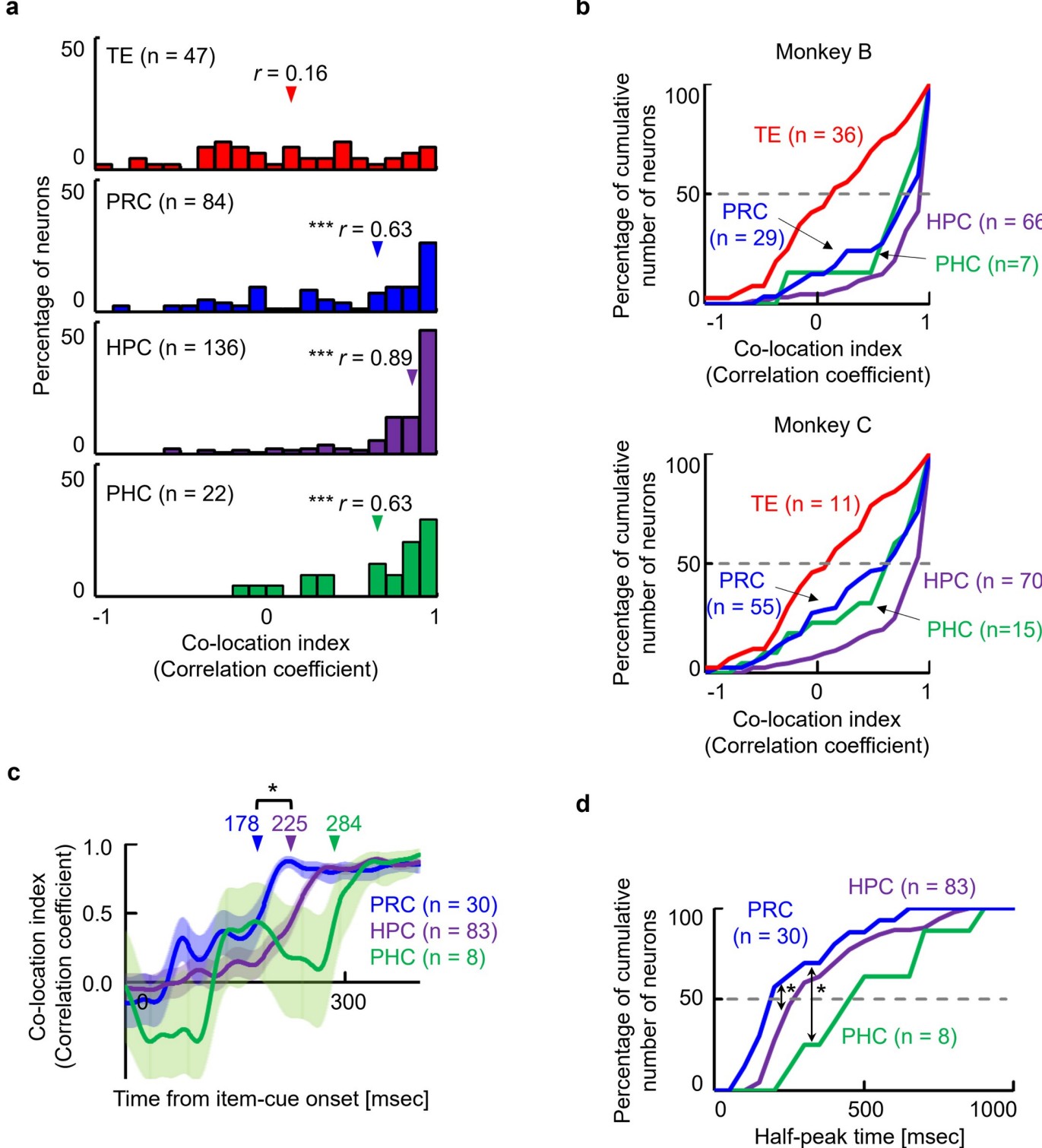

**Fig 4. Memory retrieval signal in the MTL.** (a) Distributions of co-location indices for item-cue selective neurons in each area. Red, TE ($n$ = 47); blue, PRC ($n$ = 84); purple, HPC ($n$ = 136); green, PHC ($n$ = 22). Co-location indices in all MTL areas were significantly positive ($P$ < 0.0001, Y = 0, for PRC, $P$ < 0.0001, Y = 0, for HPC, $P$ = 0.0009, Y = 0, for PHC, two-sided Wilcoxon signed-rank test) and greater than those in TE ($P$ = 0.0043, KS = 0.31 for PRC; $P$ < 0.0001, KS = 0.59 for HPC; $P$ = 0.0056, KS = 0.43 for PHC; Kolmogorov–Smirnov test). (b) Cumulative frequency histograms of co-location indices for item-cue selective neurons in 2 monkeys. For both monkeys, the HPC showed the largest co-location index (median = 0.92 and 0.84 for monkeys B and C, respectively), followed by the other 2 MTL areas (median = 0.86 and 0.35 in the PRC, 0.78 and 0.59 in the PHC for monkeys B and C, respectively). The co-location index was close to the

lowest in the TE for both monkeys B (0.17) and C (−0.04). **(c)** Time course of population-averaged co-location indices. Lines and shading, mean and SEM across the item-cue selective neurons with high co-location indices ($r > 0.8$) in MTL areas. The population-averaged co-location index increased earlier in the PRC than in the HPC ($P = 0.0382$*, two-tailed permutation test). Arrow, half-peak time. **(d)** Cumulative frequency histograms of half-peak time of co-location indices for individual neurons. Half-peak times were calculated for each item-cue selective neurons with high co-location indices ($r > 0.8$) in MTL areas. The half-peak times were shorter in the PRC than in the HPC ($P = 0.0395$*, KS = 0.29, Kolmogorov–Smirnov test) and PHC ($P = 0.0201$*, KS = 0.57). Source data are available in S1 Data. HPC, hippocampus; MTL, medial temporal lobe; PHC, parahippocampal cortex; PRC, perirhinal cortex; SEM, standard error of the mean.

$f. = 1$) (S6B Fig). We confirmed this area difference by considering the strengths ($R^2$ values) of item and target selectivity for all recorded neurons (S7 Fig). These results suggest that the neural representations during the item-cue period may be explained by activities for target locations during the choice-fixation period in the HPC and the PHC but not in the PRC.

Next, we compared the preferred locations of neurons during the item-cue period to those during the choice-fixation period. Fig 5 displays the responses of one HPC neuron that showed selective activity during both task periods. This neuron signaled co-location III during the item-cue period, whereas it signaled the bottom-left target location during the choice-fixation period (Fig 5A and 5B). The bottom-left was assigned as a target location in trials with combinations of "co-location II & 90˚," "co-location III & 0˚," and "co-location IV & −90˚" (Fig 5C). The response patterns to the target locations during the choice-fixation period and those to the co-locations during the item-cue period were positively correlated when the co-locations were assumed to be positioned relative to the 0˚ context-cue ($r = 0.99$), but not to the −90˚ context-cue ($r = -0.27$) nor the 90˚ context-cue ($r = -0.27$) (S8 Fig). These results suggest that the HPC neuron represented the co-location of the item-cue stimulus relative to a particular context-cue during the item-cue period, although the context-cue was not yet presented.

We examined this tendency in the population by conducting representational similarity analysis (RSA) using the responses from all the recorded neurons in each area ($n = 319$, PRC; $n = 456$, HPC; $n = 232$, PHC) because only a few neurons showed task-related activity in both task periods in the PRC ($n = 8$) and the PHC ($n = 6$). We calculated a correlation coefficient between the $N$-dimensional population vector for the response to each co-location during the item-cue period and that of the response to each target location during the choice-fixation period in each area (see **Materials and methods** and S9 Fig). "$N$" was the number of the recorded neurons in each area. According to 3 orientations (−90˚, 0˚, or +90˚) of the map image, co-locations were combined with particular target locations (e.g., "co-location I" and "top-left" in −90˚ orientation) to form a total of 4 combinations in each orientation. The average of the correlations across the 4 combinations showed a significantly positive value in the HPC ($r = 0.14$ and $P < 0.0001$, two-tailed permutation test) and the PHC ($r = 0.18$, $P < 0.0001$) only when assuming the co-locations on a map image with 0˚ orientation (Fig 6). These results suggest that the co-location of the item-cue was represented relative to the 0˚ map image before the presentation of the context-cue in the HPC and the PHC. The selective conjunction of the co-locations and the 0˚ map image might be due to the training history in which the monkeys had learned the ILA under the context-cue with 0˚ orientation during the initial training ("default condition") [19], although we do not exclude a possibility that the default condition might be just due to the symmetric property in orientation of the context-cue between −90˚ and 90˚. In contrast to the HPC and the PHC, a positive correlation was not found for any orientation of the map image in the PRC ($P = 0.73$, 0.77, 0.06 for −90˚, 0˚, and 90˚, respectively). The nonsignificant result of the RSA in the PRC could neither be explained by the sample size nor the strengths of task-related signals because they were even larger in the PRC than in the PHC ($n = 319$ in PRC versus 232 in PHC, sample size; $r = 0.63$ versus 0.63, co-location index; 26.3% × 6.0% versus 9.5% × 10.8%, item-cue × target location selective neurons). Together, the comparison of the response patterns between the item-cue and choice-

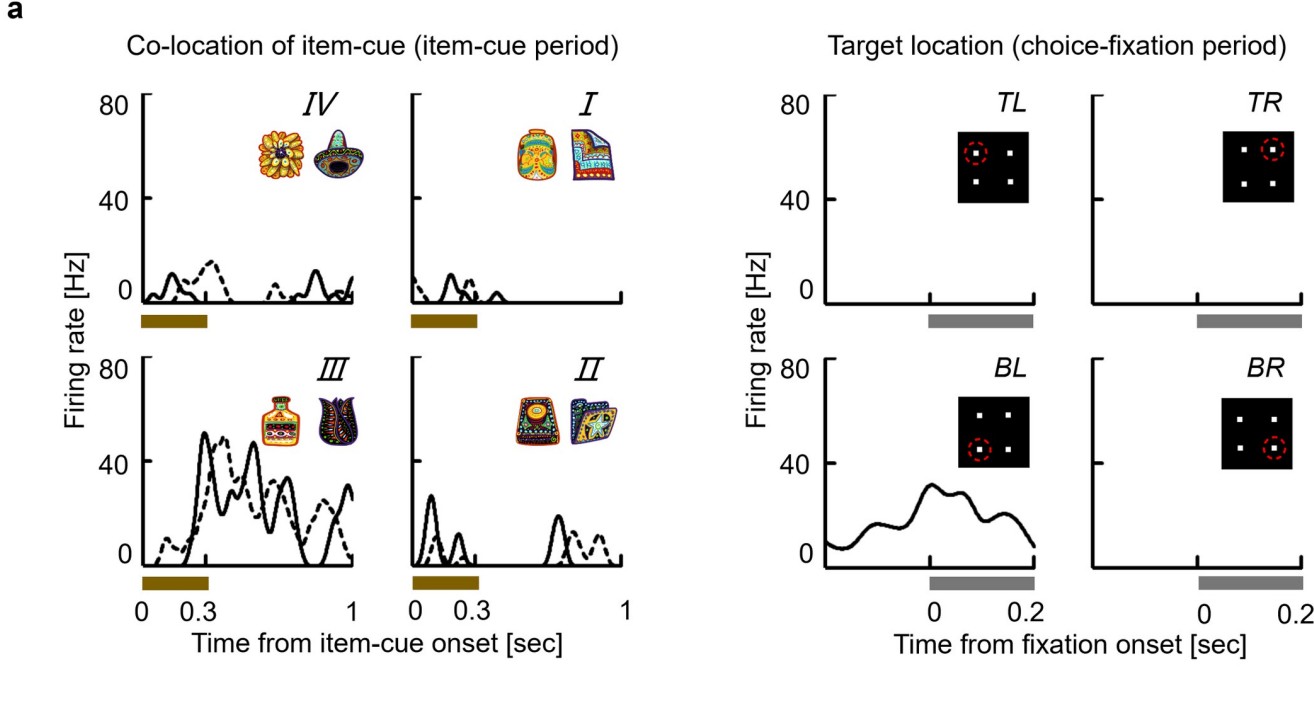

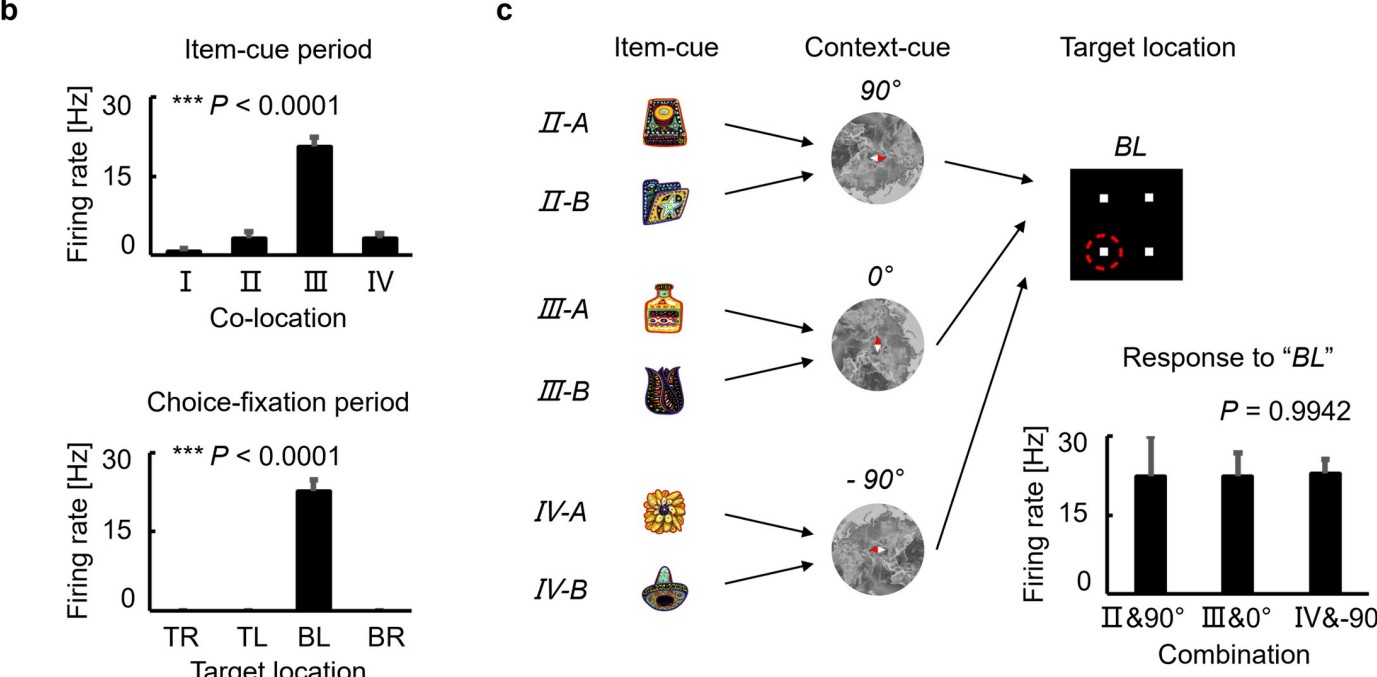

**Fig 5. An HPC neuron exhibiting both item-cue selectivity and target selectivity. (a)** Lines, SDFs for each item-cue during the item-cue period (left) or SDF for each target location during the choice-fixation period (right). I–IV, co-location I–IV. Brown bar, presentation of the item-cue. TR, top-right; BR, bottom-right; BL, bottom-left; TL, top-left. Gray bar, fixation on the target location. **(b)** Mean responses during the item-cue period for each co-location (top) and during the choice-fixation period for each target location (bottom). Error bar, standard error. Significant selectivity for the co-locations ($P < 0.0001$***, $F_{(3, 42)} = 56.6$, one-way ANOVA) and for the target locations ($P < 0.0001$***, $F_{(3, 42)} = 129.3$, one-way ANOVA). **(c)** Combinations of item- and context-cues resulted in the bottom-left (BL) target location. Mean responses during the choice-fixation period for combinations resulted in the BL target location. Error bar, standard error. Responses were not significantly different among the combinations that resulted in the BL target location ($P = 0.9942$, $F_{(2, 8)} = 0.0058$, one-way ANOVA). The background map image was made based on an image (EMU 13) from public domain, "USGS" (https://www.usgs.gov/media/images/emu-13). Source data are available in S1 Data. HPC, hippocampus; SDF, spike density function.

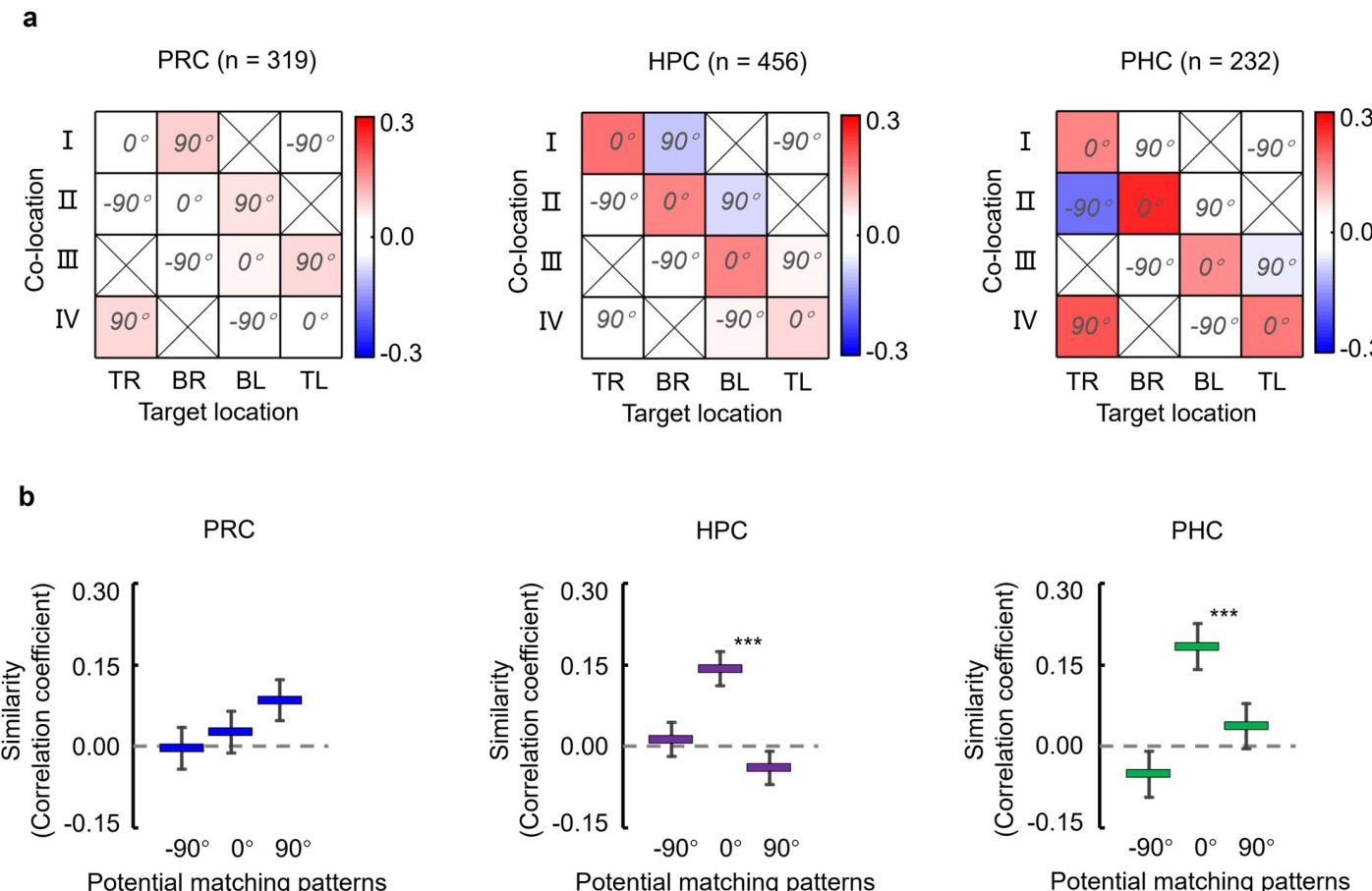

**Fig 6. Representational similarity between co-locations and target locations. (a)** Correlation matrices between population vectors for co-locations of item-cues and those for target locations in each MTL area. The population vectors consist of activity from the recorded neurons in each area (PRC, $n = 319$; HPC, $n = 456$; PHC, $n = 232$). TR, top-right; BR, bottom-right; BL, bottom-left; TL, top-left. **(b)** Representational similarity between co-locations and target locations was estimated as a mean value of correlation coefficients for 4 combinations of co-locations and target locations corresponding to each of −90˚, 0˚, and 90˚ map image conditions. Error bar, standard deviation. Asterisk indicates results of two-tailed permutation test: $P < 0.0001***$. Source data are available in S1 Data. HPC, hippocampus; MTL, medial temporal lobe; PHC, parahippocampal cortex; PRC, perirhinal cortex.

fixation periods suggests that the HPC and the PHC represent the co-locations of the item-cue stimuli in relation to a particular spatial context. Conversely, the PRC might represent co-locations irrespective of their relationship with any particular spatial context.

## Discussion

Using the same ILA task, we previously presented the retrieved co-location information represented in the HPC [19]. However, it remains unclear whether the co-location information was retrieved in the HPC or whether the HPC received a retrieval signal from other upstream brain areas. To address this problem, the present study examined the parahippocampal cortical areas, including the PRC and PHC, which provide signal inputs to the HPC. In addition to the HPC, item-cue selective neurons in the PRC and the PHC showed significantly correlated responses to pairs of co-locating items that the monkeys had learned to associate with the same co-locations in advance (Fig 4A and 4B). Among these 3 areas, correlated responses first appeared in the PRC (Fig 4C and 4D). In contrast to the HPC and parahippocampal cortical areas, correlated responses were not found in TE, which reportedly provides the PRC with

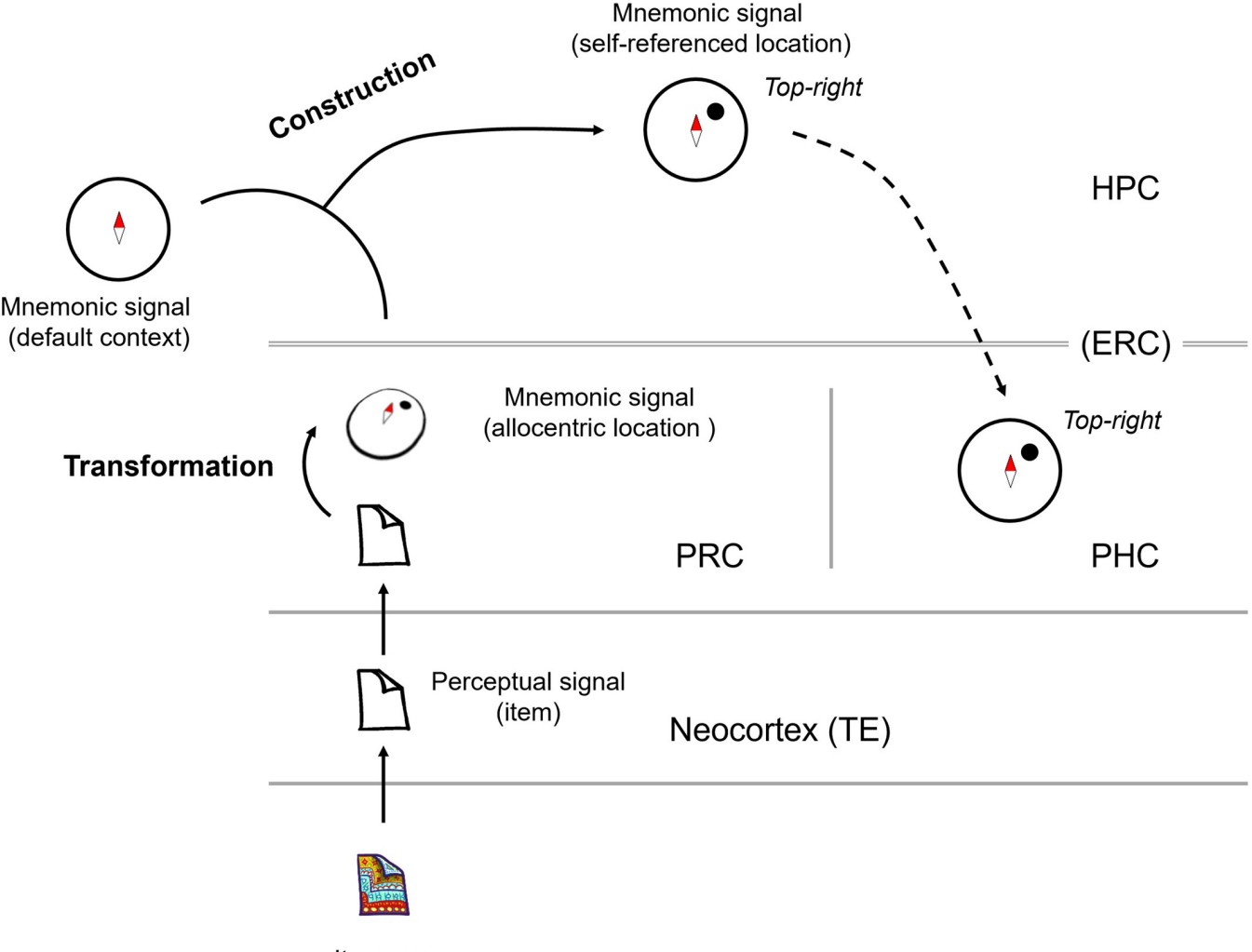

**Fig 7. Two-stage recall model of semantic memory.** Schematic diagram of neuronal signals in the MTL during the item-cue period in the ILA task. Perceptual signal of the item-cue transmits from the neocortex to the PRC, in which the item information would be transformed to the mnemonically linked location information, which is represented in an allocentric manner. The retrieval signal would transmit (via the ERC) to the HPC, in which the allocentric location would be combined with the default context to construct the self-reference representation of the retrieved location. The self-referenced location signal would spread from the HPC (via ERC) to PHC. HPC, hippocampus; ILA, item-location associative; MTL, medial temporal lobe; PHC, parahippocampal cortex; PRC, perirhinal cortex.

perceptual information of visual objects [25,26]. These results suggest that the PRC was first involved in the memory retrieval process before the HPC. We also examined the representation property of the co-location information in each MTL area and found that the retrieved co-location information was represented in relation to a particular spatial context (i.e., 0˚ of the map image) in the HPC and the PHC but not in the PRC (Fig 6). These results suggest involvements of both PRC and HPC in remembering item-location associative memory that was learned as common information across trials with different context-cues, choice responses, and reward outcomes (Fig 7).

In this study, the effect of item-location associative memory was first assessed by correlated responses to pairs of co-locating items. Similar to the present results, an increase in correlated responses to pairs of items from TE to the PRC was also found in previous studies using the pair-association paradigm [27,34,35], which examined a direct association between items

[36,37]. Using the ILA task, this study indicated that the PRC also served as an indirect linkage between items via their co-locations on the map image. In contrast to the significant involvements of the PRC in both memory paradigms, a small but significant correlation effect in TE was found only in the pair-association paradigm [27,38]. In the ILA paradigm, the lack of an association effect in TE might be explained by the unwanted recall of co-locating items from item-cues because the retrieved location information signaled forward from the PRC to HPC. Conversely, the PRC provided TE with a feedback signal to substantiate the item–item association effect in the pair-association paradigm [26,38]. Secondly, to examine the effect of item-location associative memory, we compared activities during the item-cue period with those during the choice-fixation period. This study showed that neurons in the MTL areas exhibited selective responses to target locations where the monkeys positioned their gaze for choice. The target-selective responses during the choice-fixation period could be explained by locations relative to "multiple spatial reference frames" [39], including head-center, external landmarks (e.g., computer screen), and the to-be-retained context-cue in mind, which could be represented in either the head-center or external landmark frame. This ambiguity makes it difficult to distinguish the target-selective activity in the self-referenced frame from that in the allocentric frame. However, the gaze-position-dependent activity during the choice-fixation period still seems to reflect the first-person perspective because it depended on what the subjects perceived at that time.

Our previous study using the same ILA task reported that the HPC showed target-selective activity during the context-cue period, during which the animal maintained fixation at the center of the display [19]. However, substantial number of neurons in the HPC exhibited the item-cue (co-location) and the context-cue information in addition to the target information. The intermingled task-related information during the context-cue period contrasts with the exclusive representation of the target-selective information during the choice-fixation period in the present study. This tendency was confirmed in the other MTL areas (S10 Fig). Taken together, the activity during the choice-fixation period would be more appropriate than those during the context-cue period as a reference to examine the "first-person perspective" effect on the activity during the item-cue period, particularly when we conduct population-coding data analysis such as the RSA, which includes all recorded neurons in each area. Therefore, to examine the first-person perspective effect on the retrieved location information during the item-cue period, the comparison with the choice-fixation period seems to be the most appropriate in this study, although it may not yet be a direct measure of the first-person perspective effect. A future study should assess the effect by developing a new behavioral paradigm, in which subjects report their mental imagery during a memory task, and directly compare neuronal activity with the subjective memory content that they directly report at that time [40].

The item-location associative memory isolated from a particular spatial context (e.g., the map image with 0˚ orientation) in the PRC is consistent with previous literature suggesting its involvement in semantic memory [25,26,41–44] as well as the allocentric representations of semantic memory [4,5]. The isolation from the particular context may also explain the previous studies suggesting a selective involvement of the PRC in familiarity rather than in recollection [13,45]. The reactivation of item-location associative memory in the PRC may induce familiarity [46], but the reactivated mnemonic information in the PRC may not provide a first-person perspective that could be reexperienced in mind with a particular spatial context [4,6,47]. Therefore, the location information in the PRC may not be directly coupled with mental imagery as recollection. The poor access of the reactivated item-location associative memory to mental imagery may also explain previous studies suggesting the exclusive processing of object-related information in the PRC [12,14], although recent physiological studies, including ours, have reported space-related information represented by PRC neurons

[22,25,48–50]. Here, we hypothesize that the PRC might be involved in retrieving semantic memory even when it includes the space-related information, although its reinstated spatial contents may not be directly experienced in the mind because of the allocentric representation of semantic memory [4].

In contrast to the PRC, the retrieved location was represented in relation to a particular spatial context (0˚ orientation of the map image) in the HPC and the PHC. An accompaniment of a particular spatial context is consistent with previous studies suggesting the involvement of the HPC in semantic recall with a vivid mental image [51–53], which presumably involves concrete representations from the first-person perspective. Of the 2 areas, the HPC showed a much larger retrieval signal than the PHC, which was estimated by the proportions of the item-cue selective neurons (29.8% in the HPC versus 9.5% in the PHC) and their co-location indices (median $r$-value, 0.89 versus 0.63) (Figs 2B and 4A and 4B). Moreover, the time courses of the co-location index elevated earlier in the HPC than in the PHC (half-peak time, 225 ms versus 284 ms) (Fig 4C and 4d), which may imply a transmission of the retrieved information from the HPC to the PHC. Together, it may be reasonable to deduce a critical involvement of the HPC rather than the PHC in accompanying the semantic-like item-location associative memory with a particular spatial context to represent the memory component in the first-person perspective. A future study should address whether the PHC just serves as a relay center between the HPC and the dorsal pathway [18,54] or processes the retrieval signal from the HPC. Because we could not find a qualitative difference in the retrieval signal between the PHC and HPC, it might be important to devise a new research paradigm to reveal the role of PHC.

Representations of the retrieved location in the default context appeared inconsistent with the definition of semantic memory, which should be formed in an allocentric manner [1,4,55], because the retrieved location was related to only 1 particular spatial context. However, after the context-cue presentation, the retrieved location was fitted to a spatial context given by the context-cue in the HPC [19]. In short, the retrieved location can be represented in the self-referenced frame among multiple spatial contexts, including the default context. However, the fitness of the self-referenced location at any spatial context amounts to indicate the allocentric location of the item-cue on the background map [56]. Another possibility might be that retrieval during the item-cue period reflected an episode that the animals experienced in a particular single trial during the training. This episodic-like memory might match the presence of the particular spatial context. However, it seems unlikely for the animals to recall the episode of a particular single trail, because the animals had experienced trials with the same item-cue and context-cue many times before. In addition, if the animals should have recalled the episodic-like memory, the item-cue would remind a subject of subsequent task events in a to-be-retrieved trial, such as the appearance of a context-cue, disappearance of a fixation dot (small white square), appearance of choice-fixation dots (four small white squares), choice response, and presence/absence of reward delivery in addition to the co-location information that was not shown at least before the choice response. Moreover, the recall of episodic memory might also reinstate contexts including information of preceding and succeeding trials. Together, episodic-like memory would provide the animals with plenty of unnecessary information to solve each trial of the ILA task. Therefore, it is reasonable to consider that the item-location associative memory investigated in this study could be classified as a semantic-like memory rather than an episodic-like memory, although some neural mechanisms might be shared between the 2 types of memory to represent the retrieved memory content in its default context.

One remaining question is how the retrieved location was accompanied by a particular spatial context in the absence of the map image. Using the same ILA task, we previously showed a constructive process in which the HPC neurons fitted the retrieved location to the spatial

context given by the context-cue [19]. Similarly, we hypothesize that before the context-cue presentation, HPC neurons combined item-location associative memory and default context to represent item-location from the first-person perspective (Fig 7). Considering the anatomical hierarchy of the PRC linking the ventral pathway and the HPC for object-related information [18,25,57], and the earlier rise of the co-location index in the PRC than in the HPC after the item-cue presentation, it seems reasonable to interpret that the HPC neurons received the allocentric item-location from the PRC via the ERC and fit it to the default context in the absence of the map image (Fig 7). Based on the present investigation of semantic-like memory in nonhuman primates, we propose a two-stage recall process for semantic memory. In this model, the PRC and HPC would contribute to the recall of semantic memory in distinct but complementary manners; the PRC might serve for semantic recall in the allocentric frame, while the HPC might serve for constructing the first-person perspective [56].

## Materials and methods

### Ethics statements

Animals were maintained on a 12-h light/dark cycle with a room temperature range of 18 to 29°C and a humidity range of 30% to 70%. Animal health and welfare was monitored daily. Environmental enrichment and feeding were in accordance with the National Institutes of Health (NIH) Guide for the Care and Use of Laboratory Animals and the Association for Assessment and Accreditation of Laboratory Animal Care (AAALAC) guidelines. All experimental procedures were approved by the Institutional Animal Care and Use Committee (IACUC) of Peking University (license: Psych-YujiNaya-1). Animals will be euthanized after all experiments by sedation followed by intravenous injection of sodium pentobarbital in accordance with the AVMA Guidelines for the Euthanasia of Animals.

### Experimental design

**Subjects.** The subjects were 2 adult male rhesus monkeys (*Macaca mulatta*; 6.0 to 9.0 kg).

### Behavioral task

Two monkeys were trained on an ILA task (Fig 1) and performed both the training and recording sessions under dim lights. The task was initiated by the monkey fixating on a white square (0.4° visual angle) in the center of a display for 0.5 s. Eye position was monitored by an infrared digital camera with a 120-Hz sampling frequency (ETL-200, ISCAN). Subsequently, an item-cue (diameter, 3.4°) and context-cue (diameter, 28.5°) were sequentially presented for 0.3 s each with a 0.7-s interval. After a 0.7-s delay interval, 4 white squares (0.4°) were presented as the choice stimuli an equidistance from the center (6°). One of the squares was a target, whereas the other 3 were distracters. The target was determined using a combination of the item-cue and the context-cue stimuli. The monkeys were required to saccade to one of the 4 squares within 0.5 s and to fixate on it (typically less than 2° from the target center) for 0.2 s. If they made the correct choice, 4 to 8 drops of water were given as a reward, as well as visual feedback showing the associated location of the item-cue on the context-cue (S1 Fig). When the monkeys failed to maintain their fixation or made an incorrect choice, the trial was terminated without reward or visual feedback, followed by an inter-trial interval. The monkeys were trained to associate 2 sets of 4 visual stimuli (item-cues) with 4 particular locations relative to the context-cue image that was presented at a tilt (with an orientation from −90° to 90°) before the recording session commenced. We first trained monkeys to learn the task rule of the ILA task using a preliminary stimulus set (monochromatic simple-shaped objects [e.g., cross,

heart] as item-cue stimuli and a large disk with 4 monochrome colors in individual quadrants as the context map stimulus) on a touchscreen (M1700SS, 3M) [19]. Then, the ILA task with the main stimulus set was also trained using a touchscreen (S2 Fig). To prevent the monkeys from learning to associate each combination of the item-cue and context-cue with a particular target location, the orientation of the map image was randomized at a step of 0.1˚, which increased the number of combinations (8 × 1,800) and would make it difficult for the monkeys to learn all the associations among the item-cues, context-cues, and target locations directly. During the recording session, the item-cue was pseudorandomly chosen from the 8 well-learned visual items, and the orientation of the context-cue was pseudorandomly chosen from among 5 orientations (−90˚, −45˚, 0˚, 45˚, and 90˚) in each trial, resulting in 40 (8 × 5) different configuration patterns. We trained both monkeys using the same stimuli but with different ILA patterns. All the stimulus images were created using Photoshop (Adobe). Both monkeys performed the task correctly (chance level = 25%) at rates of 80.5% ± 8.4% (mean ± standard deviation; Monkey B, $n$ = 435 recording sessions) and 95.1% ± 5.8% (Monkey C, $n$ = 416 recording sessions).

## Electrophysiological recording

The monkeys were each implanted with a head post and a recording chamber under aseptic conditions, using isoflurane anesthesia, following the initial behavioral training. We used a 16-channel vector array microprobe (V1 X 16-Edge; NeuroNexus) or a single-wire tungsten microelectrode (Alpha Omega) to record single-unit activity, which was advanced into the brain using a hydraulic Microdrive (MO-97A; Narishige) [32]. The microelectrode was inserted through a stainless-steel guide tube positioned in a customized grid system in the recording chamber. Neuronal signals for single units were collected (low-pass, 6 kHz; high-pass, 200 Hz) and digitized (40 kHz) (AlphaLab SnR Stimulation and Recording System, Alpha Omega). These signals were sorted using an offline sorter (Plexon). An average of 87 trials were tested for each neuron ($n$ = 1,175). The placement of the microelectrodes into the target areas was guided by individual brain atlases from MRI scans (3T, Siemens). Individual brain atlases were also constructed based on the electrophysiological properties around the tip of the electrode (e.g., gray matter, white matter, sulcus, lateral ventricle, and bottom of the brain). The recording sites were estimated by combining the individual MRI and physiological atlases [58].

The recording sites covered 3 to 19 mm anterior to the interaural line (monkey B, left and right hemisphere; monkey C, right hemisphere; Fig 2A). The recording sites in the HPC appeared to cover all its subdivisions (i.e., the dentate gyrus, CA3, CA1, and subicular complex) [19]. The recording sites in the PRC appeared to cover area 36 from the fundus of the rhinal sulcus to the medial lip of the anterior middle temporal sulcus (amts) [24,27,59]. The recording sites in the PHC appeared to cover area TF [60]. The recording sites in the TE were limited to the ventral area, including both banks of the amts.

## Statistical analysis

All neuronal data were analyzed using MATLAB R2020a (MathWorks) with custom-written programs, including the statistics toolbox. The permutation tests were performed with 10,000 shuffle iterations.

## Classification of task-related neurons

For the item-cue period, we calculated the mean firing rates of 8 consecutive 300-ms time-bins moving in 100-ms steps, from 0 to 1,000-ms after item-cue onset, across all correct trials. We

evaluated the effects of "item" for each neuron using one-way ANOVA with the 8 item-cue stimuli as the main factor ($P < 0.01$, Bonferroni correction for eight-analysis time windows). We refer to neurons with significant item effects during any of the eight-analysis time windows as item-cue selective neurons. For the choice-fixation period, we calculated the mean firing rates of the 200-ms choice-fixation period when the subject fixated on the target location across all correct trials with −90˚, 0˚, and 90˚ context-cues. We evaluated the effects of "item," "context," and "target location" for each neuron by using three-way ANOVA with the 8 item-cue stimuli, 3 context-cue orientations, and 4 target locations as main factors ($P < 0.01$).

## Display of example neurons

To show the activity time course for an example of an item-selective neuron, a spike density function (SDF) was calculated using only correct trials and was smoothed using a Gaussian kernel with a sigma of 20 ms (Figs 3A and S3). The deviation in the response to each item-cue was evaluated using bootstrap resampling. For each correct trial, we smoothed the response across time using a Gaussian kernel with a sigma of 20 ms. For each item-cue, bootstrap data samples and the time courses of mean responses were generated 10,000 times. A 90% confidence interval of the 10,000 bootstraps was plotted. Only SDFs were shown during the item-cue and choice-fixation periods as an example of item-cue selective neurons exhibiting target-selective responses during the choice-fixation period (Fig 5A).

## Item-location associative effect during the item-cue period

We examined the item-location associative effect on item-cue selective activity by calculating the Pearson correlation coefficients between the responses to co-locating items. For each item-cue selective neuron, we calculated mean firing rates for each 300-ms time-bin, moving by 100 ms during the item-cue period for each correct trial (8 bins in total). We subsequently performed a one-way ANOVA and calculated the grand mean for each item stimulus across correct trials for each bin. For the bins that showed a significant ($P < 0.01$, Bonferroni correction for eight-analysis time windows) item effect, we calculated the correlation coefficients between the mean firing rates of the co-locating items. After which, we averaged $Z$-transformed values of the correlation coefficients across time-bins for each neuron. Finally, the average value was transformed into $r$ values (i.e., the co-location index).

To examine the time courses of the retrieval signal in each area, we calculated the correlation coefficients for each 100-ms time-bin moving by 1 ms during the item-cue period for each item-cue selective neuron with high correlation coefficients. We then averaged the correlation coefficients using $Z$-transformation for each time-bin across neurons for each area. The half-peak time was defined as the time from the item-cue onset to the instant when the population-averaged correlation coefficient ($r$-value) in each area, reached 50% of its peak rise from 0 (Fig 4C). We also detected the half-peak time of the correlation coefficients for individual neurons (Fig 4D). For this purpose, we eliminated noise components that were inevitably contained because of the small degree of freedom in the correlation coefficient (4 pairs, $d.f. = 2$) by employing the trial resamples and averaging the results of their resamples for each neuron as follows (S5 Fig). We first averaged responses during the 60- to 1,000-ms period from item-cue onset in each trial and calculated a grand mean across trials for each of the 4 items of 1 set (e.g., set A) as a template vector $X$ (e.g., $X = [f_{AI}, f_{AII}, f_{AIII}, f_{AIV}]$). A temporally variable vector $Y$ containing responses to each of the 4 items from the other set (e.g., set B) at each time "$t$" was generated by randomly resampling 1 trial for each of the 4 items (e.g., $Y(s,t) = [f_{BI}(s,t), f_{BII}(s,t), f_{BIII}(s,t), f_{BIV}(s,t)], s = 1$ to $200$). "$f_{BI}(s,t)$" indicates the average response during 100-ms time-bin with time point "$t$" in the middle of it in a trial with "item I-B" as the item-cue that was

resampled at *s*-th resample. The Pearson correlation coefficients of vectors **X** and **Y** were calculated for each resample in each time-bin (e.g., $r_{AB}(s,t) \sim [X, Y(s,t)]$). Either of set A and set B was chosen for the template vector, and a total of 400 ($200 \times 2$) resamples were conducted for each cell. The correlation coefficients at each time-bin were transformed into *Z* values and then averaged. The averaged *Z* values were retransformed into *r* values to generate the time course of the correlation coefficients. The half-peak time was defined in the same manner as that of the population-averaged correlation coefficient.

### Relationships between retrieved co-locations and target locations

The RSA was performed using data from the correct trials (with −90˚, 0˚, and 90˚ context-cues) of all the recorded neurons. For each neuron, we first averaged the responses during the item-cue period (60 to 1,000 ms period from the item-cue onset) in each trial and calculated a grand mean across trials for each of the 4 co-locations. For each area, an *N*-dimensional population vector was prepared for each co-location. "*N*" was the number of the recorded neurons in the area. The *i*-th element of the population vector indicated the average firing rate of the *i*-th neuron for the co-location. Four population vectors were generated for the responses during the item-cue period. We also prepared for 4 population vectors indicating the responses to the 4 target locations during the choice-fixation period (0 to 200 ms from the onset of fixation on the target location) in each area. Each of the −90˚, 0˚, and 90˚ context-cues was assigned 4 particular combinations between co-locations and target locations (S9 Fig). We calculated the Pearson correlation coefficients between the population vectors for a co-location and a corresponding target location in each combination (e.g., "co-location I" and "top-left" for −90˚). After the Z-transformation, the correlation coefficients were averaged across the 4 combinations and reversed to *r* value.

### Supporting information

**S1 Table. Numbers of task-related neurons.** Numbers of neurons showing item-cue effect during the item-cue period ($P < 0.01$, one-way ANOVA) and those showing the item-cue, context, and target effects during the choice-fixation period ($P < 0.01$, three-way ANOVA). "Item-cue" indicates an item effect during the item-cue period or choice-fixation period. "Context" indicates a context effect during the choice-fixation period. "Target" indicates a target location effect during the choice-fixation period. Source data are available in S1 Data. (DOCX)

**S1 Fig. Example of a feedback image.** The visual feedback image of 1 example trial was magnified. Scale bar for feedback image, 5˚ visual angle. The background map image was made based on an image (EMU 13) from public domain, "USGS" (https://www.usgs.gov/media/images/emu-13). (TIF)

**S2 Fig. Example trials of the item-location association task during the training phase using a touchscreen.** An item-cue and a context-cue were sequentially presented in each trial. The item-cue was chosen randomly from the 8 visual items, and the context-cue was presented with a randomly chosen orientation from −90˚ to 90˚ in a 0.1˚ step. Monkeys had to make a choice by touching the target location (red dashed circle) according to the 2 cues. A successful trial was rewarded with juice paired with feedback showing the associated location of the item-cue on the context-cue. Relative sizes of the stimuli were magnified for display purposes. The background map image was made based on an image (EMU 13) from public domain, "USGS"

(https://www.usgs.gov/media/images/emu-13).
(TIF)

**S3 Fig. Example neurons showing the item-cue selective activities during the item-cue period.** Following a similar format as Fig 3A. Solid lines and dashed lines indicate SDFs in trials with item-cues from the stimulus sets A and B, respectively. Dark and light gray shading, 90% confidence interval of 10,000 bootstraps for the stimulus sets A and B, respectively. Black and gray dots, raster plots for the stimulus sets A and B, respectively. Brown bar, presentation of the item-cue. **(a)** An example neuron from TE. The neuron showed the item-cue selective activities but not the co-location effect. $P < 0.0001$, $F(7,115) = 153.41$, one-way ANOVA. Co-location index $r = 0.16$, Pearson correlation; $P = 0.87$, two-tailed permutation test. **(b)** An example neuron from the HPC. The neuron showed the co-location effect on the item-cue selective activities. $P < 0.0001$, $F(7,70) = 45.09$, one-way ANOVA. Co-location index $r = 0.99$, Pearson correlation; $P = 0.0002$, two-tailed permutation test. **(c)** An example neuron from the PHC. The neuron showed the co-location effect on the item-cue selective activities. $P < 0.0001$, $F(7,57) = 15.74$, one-way ANOVA. Co-location index $r = 0.99$, Pearson correlation; $P = 0.0004$, two-tailed permutation test. Source data are available in S1 Data. (TIF)

**S4 Fig. Time courses of co-location index using different neuron screening criteria. (a)** For the item-cue selective neurons with high co-location indices ($r > 0.7$) in PRC (blue, $n = 38$), HPC (purple, $n = 102$), and PHC (green, $n = 9$). The formats are the same as those in Fig 4C. The co-location index increased earlier in the PRC than in the HPC ($P = 0.0114$*, two-tailed permutation test). **(b)** For item-cue selective neurons with significant co-location effect ($P < 0.05$, two-tailed permutation test) in PRC (blue, $n = 15$), HPC (purple, $n = 60$), and PHC (green, $n = 5$). The formats are the same as those in Fig 4C. Source data are available in S1 Data. (TIF)

**S5 Fig. A schematic illustration of a procedure detecting half-peak time for individual neurons. (a)** PSTHs and time course of the co-location index of an example item-cue selective neuron with a high colocation index (0.97). The neuron showed a large value of the co-location index even before the item-cue presentation. This type of noise tends to occur because of the small degree of freedom in the correlation coefficient. **(b)** To eliminate the inevitable noise component, we employed the trial resample and average method within an individual neuron. Template vector $X$, average responses across trials to each of the 4 items of 1 set during the 60- to 1,000-ms period from item-cue onset. Either set A or set B was chosen as the template vector alternatively. Temporally variable vector $Y(s,t)$, responses of 1 trial at $s$-th resample to each of the 4 items of the other set at each time "$t$"; $t$, the center of 100-ms time-bin moving by 1 ms during the item-cue period. For each set, vector $Y$ was generated 200 times by resampling trials. The dimensions of the 2 vectors were both 4. $r_{AB}(s,t)$, the correlation coefficient between $X$ and $Y$ at $s$-th resample at time $t$ wherein the template vector was generated from set A and the temporally variable vector was generated from set B, $s = 1$ to 200. $r(t)$, the average correlation coefficient at time $t$. The half-peak time of each neuron was calculated from the time course of the average correlation coefficient. (TIF)

**S6 Fig. Task-related signals during the item-cue period and choice-fixation period. (a)** Percentages of item-cue, context-cue, and target-selective neurons out of the recorded neurons during the choice-fixation period. PRC, $n = 319$; HPC, $n = 456$; PHC, $n = 232$. Dashed line, 1.0% chance level. Asterisks indicate results of one-tailed binomial test (probability of null

hypothesis = 1.0%): $P = 0.0432^*$ for context-cue selective neurons in PRC; $P = 0.0003^{**}$ for context-cue selective neurons in the HPC; $P < 0.0001^{***}$ for target-selective neurons in each area. **(b)** Percentage of target-selective neurons during choice-fixation period out of item-cue selective neurons (PRC, $n = 84$; HPC, $n = 136$; PHC, $n = 22$). Asterisks indicate results of a $\chi$-square test: $P = 0.0170^*$, $\chi^2 = 5.7$, $d.f. = 1$; $P = 0.0087^{**}$, $\chi^2 = 6.88$, $d.f. = 1$. Source data are available in S1 Data.
(TIF)

**S7 Fig. Strengths of item-cue selectivity and target selectivity in the MTL.** Item-cue selectivity index, the $R^2$ value of item effect from one-way ANOVA test during the item-cue period. Target selectivity index, the $R^2$ value of target location effect from three-way ANOVA test during the choice-fixation period. Each dot indicates 1 neuron. The significant correlations between the item-cue selectivity index and target selectivity index were found in the HPC ($P < 0.0001^{***}$, two-tailed permutation test) and PHC ($P = 0.0084^{**}$). Source data are available in S1 Data.
(TIF)

**S8 Fig. Correspondences between co-locations and target locations under 3 potential patterns.** Explanatory pattern, the co-locations were assumedly positioned relative to a −90˚, 0˚, or 90˚ context-cue. *r*, the similarity between response patterns to the co-locations during the item-cue period and those to the target locations during the choice-fixation period. I-IV, co-location I-IV. TR, top-right; BR, bottom-right; BL, bottom-left; TL, top-left. Asterisks indicate the results of a two-tailed permutation test: $P = 0.0074^{**}$. Source data are available in S1 Data.
(TIF)

**S9 Fig. A schematic illustration of RSA.** *X(co-location)*, *N*-dimensional population vector for each co-location. *f(co-location*, *i)*, the average firing rate of the *i*-th neuron for the co-location during the item-cue period. *N*, number of recorded neurons in the area. *Y(target)*, population vector for each target location. *f(target, i)*, the average firing rate of the *i*-th neuron for the target location during the choice-fixation period. *r*, Pearson correlation coefficient between *X(co-location)* and *Y(target)*. *−90°, 0°, or 90°*, explanatory pattern, the co-locations were assumedly positioned relative to a −90˚, 0˚, or 90˚ context-cue. The background map image was made based on an image (EMU 13) from public domain, "USGS" (https://www.usgs.gov/media/images/emu-13).
(TIF)

**S10 Fig. Task-related signals during the item-cue, context-cue and choice-fixation periods.** Percentages of item-cue (top row), context-cue (middle row), and target location (bottom row) selective neurons out of the recorded neurons during 3 periods. Left column, item-cue period, 0–1,000 ms from item-cue onset, three-way ANOVA ($P < 0.01$, *Bonferroni correction* for eight-analysis time windows). Middle column, context-cue period, 0–1,000 ms from context-cue onset, three-way ANOVA ($P < 0.01$, *Bonferroni correction* for eight-analysis time windows). Right column, choice-fixation period, 0–200 ms from choice-fixation onset, three-way ANOVA ($P < 0.01$). "Item," "context," and "target" effects for each neuron were evaluated across all correct trials with −90˚, 0˚, and 90˚ context-cues. PRC, $n = 319$; HPC, $n = 456$; PHC, $n = 232$. Dashed line, 1.0% chance level. Source data are available in S1 Data.
(TIF)

**S1 Data. Source data for the main figures, supporting figures, and supporting table.** The source data used to generate main figures, supporting figures, and supporting table are included under the file name S1 Data. Source data for each figure and table are arranged by

sheet and are labeled. The raw spike files for each neuron are available at: https://osf.io/vaet6/?view_only=2a3fb059695143178b323b6c6cbcb028.
(XLSX)

## Acknowledgments

We thank S. Xue for expert animal care. We thank J. Gao, W. Men, G. Yang, and the National Center for Protein Sciences at Peking University for assistance with MRI scanning. We thank D. Lanham for providing the source images of the main stimulus set.

## Author Contributions

**Conceptualization:** Yuji Naya.

**Data curation:** Cen Yang.

**Formal analysis:** Cen Yang.

**Funding acquisition:** Yuji Naya.

**Investigation:** Cen Yang.

**Methodology:** Yuji Naya.

**Project administration:** Yuji Naya.

**Resources:** Yuji Naya.

**Software:** Cen Yang.

**Supervision:** Yuji Naya.

**Validation:** Yuji Naya.

**Visualization:** Cen Yang.

**Writing – original draft:** Cen Yang.

**Writing – review & editing:** Cen Yang, Yuji Naya.

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
