## [Editor Report · Decision Letter 0]

2 Sep 2022

Dear Dr Naya, 

Thank you for submitting your manuscript entitled "Sequential involvements of macaque perirhinal cortex and hippocampus in semantic-like memory including spatial component" for consideration as a Research Article by PLOS Biology.

Your manuscript has now been evaluated by the PLOS Biology editorial staff, as well as by an academic editor with relevant expertise, and I am writing to let you know that we would like to send your submission out for external peer review. I also apologize for taking so long to get this initial decision out to you. Our academic editor was unexpectedly away so it took a bit of time for me to connect with them.

Before we can send your manuscript to reviewers, we will need you to complete your submission by providing the metadata that is required for full assessment. To this end, please login to Editorial Manager where you will find the paper in the 'Submissions Needing Revisions' folder on your homepage. Please click 'Revise Submission' from the Action Links and complete all additional questions in the submission questionnaire.

Once your full submission is complete, your paper will undergo a series of checks in preparation for peer review. After your manuscript has passed the checks it will be sent out for review. To provide the metadata for your submission, please Login to Editorial Manager (https://www.editorialmanager.com/pbiology) within two working days, i.e. by Sep 04 2022 11:59PM.

Kind regards,

Kris

Kris Dickson, Ph.D. (she/her)

Neurosciences Senior Editor/Section Manager

PLOS Biology

kdickson@plos.org

---

## [Decision Letter · Decision Letter 1]

31 Oct 2022

Dear Dr Naya,

Thank you for your patience while your manuscript "Sequential involvements of macaque perirhinal cortex and hippocampus in semantic-like memory including spatial component" was peer-reviewed at PLOS Biology. It has now been evaluated by the PLOS Biology editors, an Academic Editor with relevant expertise, and by several independent reviewers. 

In light of the reviews, which you will find at the end of this email, we would like to invite you to revise the work to thoroughly address the reviewers' reports, ensuring you address both their technical issues and the concerns raised regarding the current framing/interpretation of the study results.

Given the extent of revision needed, we cannot make a decision about publication until we have seen the revised manuscript and your response to the reviewers' comments. Your revised manuscript is likely to be sent for further evaluation by all or a subset of the reviewers.

**IMPORTANT - SUBMITTING YOUR REVISION**

*Re-submission Checklist*

*Published Peer Review*

*PLOS Data Policy*

*Blot and Gel Data Policy*

Sincerely,

Kris

Kris Dickson, Ph.D., (she/her)

Neurosciences Senior Editor/Section Manager

PLOS Biology

kdickson@plos.org

REVIEWS:

Reviewer's Responses to Questions

PLOS authors have the option to publish the peer review history of their article (what does this mean?). If published, this will include your full peer review and any attached files.

Reviewer #1: No

Reviewer #2: No

Reviewer #3: No

Reviewer #1: Yang and Naya report on the activity of neurons in perirhinal cortex, area TE, parahippocampal cortex and hippocampus while macaque monkeys perform a visual associative memory task with both object and spatial components. This is an interesting study that reports dissociations between perirhinal cortex (PRC) and hippocampus (HPC) encoding of task variables. Based on this information, Yang and Naya argue that the perirhinal cortex and hippocampus play different roles in semantic memory recall and/or retrieval.

This is an ambitious study that aims to understand the neural mechanisms underlying object-location memory and recall. There is a history of work on the neural basis of object-object (or paired associate) learning and recall in macaques, and of object-location memory. The present study extends that framework to consider object-spatial location associations in a novel way. Although I applaud the approach, and the findings are novel and interesting, I am less enthusiastic about the framing. I say more about this below.

In the field of learning and memory, there are many psychologists and neuroscientists, including both neurophysiologists and neuropsychologists, who simply do not believe that macaques (or other nonhuman animals) possess declarative memory. For the authors to make the assumption that macaques have declarative memory, and that it works just like human declarative memory, is overreaching. Books devoted to the exploration of the topic include (The Gap, by Suddendorf; The Evolution of Memory Systems, by Murray, Wise and Graham). While the findings of the present study are novel, I would urge the authors to present them in the context of associative learning. It is certainly fair enough to make the case in the Discussion that the present findings align with aspects of semantic memory in humans, but even that requires some explanation for the emphasis on perirhinal cortex and not area TE. It is evident that semantic memory in humans centers not on perirhinal cortex, but on a more lateral portion of temporal cortex that represents features at a categorical level (L Tyler et al., J Cogn Neurosci 2013; DA Levy et al., PNAS 2004; R Insausti et al., PNAS 2013). If this is the case, how do the authors explain the lack of area TE involvement in their 'semantic' retrieval? 

Specific comments: major 

1) Please reframe the results in the context of associative learning, as opposed to the framework of human declarative memory.

2) It is somewhat surprising that area TE did not show co-location effects, which are essentially item-item associations, perhaps mediated via the map. The authors discuss this briefly in the Results section, explaining that TE is important for perception, not memory (lines 152-154). To this referee, however, the item-item paired associate learning (as exemplified by pair-coding effects) and the co-location effects are both examples of memory retrieval. There should be some discussion of the present results relative to the results of earlier work on item-item paired associated learning (e.g. S Higuchi and Miyashita, PNAS, 1996). Specifically, if area TE shows correlates of item-item retrieval in the paired-associates task, why are there no significant co-location effects in TE in the present study? This should be addressed in the Discussion.

3) The nature of the visual feedback is important for understanding how the monkeys learned the ILA task. The 'Reward and feedback' column in Figure 1C is not large enough to see the detail. In the feedback period, is the item from the item-cue period shown in the correct location of the map? If not, what is given for feedback, beyond the reward itself? If there is visual feedback, please show a larger image of feedback column (either here or in Supplemental methods) and describe in the Methods. If there is no visual feedback beyond presentation of the map at the same orientation as in context-cue period, please state this is the case. What is the feedback after an incorrect response (error)? 

Specific comments: minor

1) Line 73: the authors say the monkeys 'needed to store' the common relationship between the items and the map image. Strictly speaking, this is not the case. Monkeys can learn hundreds of specific visuomotor conditional associations - independently - even though one would have thought they learned a more general rule. The fact that the study did not use rotations beyond 90 degrees during the recording sessions suggest the monkeys had a hard time learning this task, and never quite grasped the 'semantic' aspects of it. I suggest replacing 'needed to store' with were 'encouraged to store' or a similar phrase.

2) Lines 79-80: the phrase 'positioning their gaze' is awkward. It would be better to simply say the monkeys 'made a saccade to the location corresponding to the correct quadrant on the map' or similar.

3) Lines 109-110: the phrasing 'the monkeys reported a target location which corresponded to the relative location of the co-location of the item cue' is awkward. As I understand it, the monkeys simply made a saccade to the target location corresponding to the item location on the map, taking into account the orientation of the map on each trial. In performing the task, co-location is not relevant; the monkeys may know that two items correspond to each quadrant, but that may not be important to how they perform the task. 

4) Lines 109-110: Here or in Methods, please specify the degrees of visual angle (DVA) that surrounded each target 'spot', and the amount of time the monkey needed to dwell on the target to complete the choice. 

5) Lines 143-144: the authors suggest the co-location effect is due to item-item associative memory (mediated via the map). Can the authors rule out the alternative possibility that —during the item-cue period and ensuing delay—the neurons are doing prospective coding of the saccade and/or target location on the screen? Wouldn't that also lead to a co-location effect? 

6) Lines 157-159: this sounds like 'cherry picking'. What is the rationale for analyzing only neurons with high co-location indices? Fig. S4 aside, what happens if you analyze all neurons with significant co-location effects?

7) Lines 226-231: the authors find a significant relationship between coding for co-location during the item-cue period and coding for target location during the choice-fixation period (in the hippocampus and parahippocampal cortex) only when the map is oriented in the upright position (0 degrees). They then explain this may be due to a training effect. Doesn't this present a problem with the view that this is a mechanism for semantic-like retrieval? If that were the case, wouldn't there be significant relationships for more map orientations? 

Reviewer #2: Comments on Yang and Naya PLOS Biol 2022

In this article, Yang and Naya addressed the question of which brain area is involved in the retrieval of semantic memory, particularly when it contains spatial components, and the distinct role each area plays, using an item-location association (ILA) paradigm and recording single-unit activity from four different brain areas: the area TE, the perirhinal cortex (PRC), the hippocampus (HPC), and the parahippocampal cortex (PHC), in two macaque monkeys. They found that the item-location associative effect was observed in PRC, HPC, and PHC, but not in TE, and that the effect appeared earlier in PRC than in HPC and PHC. They also found that neural representations of retrieved item-cue locations were related to those of the external space that the monkeys viewed, only in the HPC and PHC but not in the PRC. Finally, they proposed a model for semantic recall that included spatial components. These observations are novel, and the experimental procedures and data analyses are mostly sound. Additional analyses are suggested that would further bolster the authors' observations below.

 I noted that this article is essentially an extension of the authors' previous paper (Yang and Naya, PLOS Biology, 2020). The same behavioral task was used in both papers, although the task is called as "the item-location associative (ILA) paradigm" in the present paper and "the constructive memory-perception (CMP) task" in the previous paper. Data were recorded from the same monkeys in both studies. The database of HPC neurons appears the same in both papers (a total of 456 cells: 247 cells in MonkeyB and 209 cells in MonkeyC, in both papers). 

In Fig.4a, the authors showed that the co-location index was significantly high in HPC, PRC, and PHC, but not in TE. The result would become more solid if the data from individual monkeys are also shown separately.

In Fig.4b, the authors demonstrated that the population-averaged co-location indices over time from the item-cue onset have increased earlier in the PRC than in the HPC and PHC in the selected neurons with high co-location indices (r > 0.8). However, I wondered whether this analysis with the population-averaged co-location index may have masked the presence of a small population of early response-onset neurons in the HPC and PHC. Since this is one of the core observations in this study, it would be better if an index from individual neurons supports the same conclusion. For example, the time of onset when the co-location index deviates significantly from the baseline should be determined in each neuron, and the distribution of the time of onset in each MTL area should be further analyzed and compared.

In the present study, the authors analyzed the "first-person perspective effect of the retrieved location" during the choice-fixation period. However, they performed the same analysis during the "background-cue period" (i.e., Background-cue period plus Delay 2 period) with HPC neurons in their previous study (Yang and Naya, 2020). It is interesting to see the results of the same analysis of the "background-cue period" data from each MTL area (i.e., not only HPC but also PRC and PHC) just as done in their previous paper, because more neurons would have shown task-related activity in both the item-cue period and background-cue period in the PRC and PHC, than in both the item-cue period and choice-fixation period. Specifically, the representation of task-related information during the "background-cue period" should be examined by applying a three-way (item-cue, context-cue, and target) ANOVA for each neuron in each MTL area and should be displayed just as for the choice-fixation period data in the S1 table and Fig.5 of the present paper. It is interesting to examine the "Similarity of orientation tuning" and the "Matching index" in the "background-cue period" and display the results for each MTL area, as shown in Fig.4c and Fig.5c of the previous paper. The results of the RSA may also be displayed for the data from the "background-cue period," just as for those from the choice-fixation period shown in Fig.7b of the present paper.

Minor

The authors used the term, "semantic-like memory" (rather than "semantic memory"), in the full title of this paper. However, they neither provided any discussion nor cited any references for their choice of this term. Please discuss this issue and provide appropriate references.

P.12, lines 226-229: The authors suggest that the selective conjunction of the co-locations and the 0° map image might be due to the training history in which the monkeys had learned the item-location association under the context-cue with 0° orientation during the initial training ("default condition"). Do the authors have any independent evidence to support this suggestion?

P.5, line 90: some typos? 

Reviewer #3: 

In their manuscript Yang and Naya principally aim at deciphering the role of the PRC in the recall of semantic memory with a spatial component, and at identifying the sequence following which MTL areas are recruited within this framework.

Despite the few conceptual short-cuts in this manuscript, the focus of this study: identifying the neuroanatomical MTL network and mechanisms supporting semantic memory, is timely and of high relevance for a wide spread audience, the technical approach (in vivo electrophysiology in macaque) very challenging and the behavioral approach chosen (comparison of item-cue period (allocentric strategy) with the choice-fixation period (as first-person perspective strategy) intriguing. As far as I could evaluate them, analyses are elegant and sound and the demonstration of the (in)dependency of one areas or the other to the spatial/non-spatial features of the task convincing.

My main concern resides on whether the process studied in the present paper is truly semantic or semantic-like memory recall for item-location pairs and not a more episodic form of this memory which could also be recalled by the two strategies suggested (allocenric versus first person perspective). How can author rule out this is not the case? To my opinion, this is not yet convincing enough (but might become upon further clarifications and discussion of the topic). Since this point has obviously a major impact on the interpretation of the data, it should be thoroughly addressed and substantiated.

As a side point, in 'the Retrieval of item-location associative memory p9, Fig 4b' part: the main results is that the co-location index increases earlier in the PRC than the HPC and PHC. statistics supporting this claim should be included in the results section. In addition, the co-location index appears to also increase earlier in the HIP than the PHC, which is not especially expected. Please provide also these comparisons and further extend the discussion on the role of the PHC within this framework. Also, as much as I have sympathy for the limited space allocated in abstracts, a statement such as 'the standard consolidation theory suggests....while the PRC is involved in its long-term storage (i.e. semantic memory)" ...shall definitively be rephrased as to my knowledge the SCT has not made such a specific claim especially involving the PRC and/ or its involvement in semantic memory, nor did the senior author of the present manuscript in a previous study (see Naya et al, 2016: The perirhinal cortex represents both between-domain associations (e.g., item-reward, item-place and item-time) and within-domain associations (e.g., item-item) and contributes to both subcategories of declarative memory (i.e., episodic and semantic memory) in a way that is complementary with the hippocampus".

further suggestions/comments:

- some discussion parts have found their way to the results sections (example p9)- they probably should not; Figures: Fig 4a and b) add names of the brain areas directly on the graph, legends: spell out the message to take home in each legend, run a spell check for typos

---

## [Decision Letter · Decision Letter 2]

23 Feb 2023

Dear Dr Naya,

Thank you for your patience while we considered your revised manuscript "Sequential involvements of macaque perirhinal cortex and hippocampus in semantic-like item-location associative memory" for publication as a Research Article at PLOS Biology. I am now handling your manuscript since Kris Dickson has moved on from PLOS Biology. This revised version of your manuscript has been evaluated by the PLOS Biology editors, the Academic Editor and the original reviewers.

Based on the reviews, we are likely to accept this manuscript for publication, provided you satisfactorily address the remaining points raised by the reviewers. Please also make sure to address the following data and other policy-related requests that I have listed below (A-F):

(A)We would like to suggest that the following modification the title:

"Sequential involvement of the perirhinal cortex and hippocampus in the recall of item-location associative memory in macaques”

(B) In the Methods section of the manuscript, please include the specific approval number issued by your IACUC/ethics committee to conduct the study.

In addition, please provide additional details regarding housing conditions, feeding regimens, environmental enrichment, and all relevant steps taken to alleviate suffering of the animals (anesthesia, analgesia, details about humane endpoints, euthanasia, etc.). Also indicate how often animal care staff monitored the health and well-being of the animals and the criteria used to make such assessments. Lastly, specify the disposition of animals at the end of the study (e.g. euthanasia, returned to home colony, etc.). If animals were euthanized following the study, please provide the method of sacrifice. 

(C) Thank you for already providing the source data for the figures presented in the manuscript (S1_Data). However, we note that the underlying data for the following figures are missing from the file:

Figure 2B, 3A, S3A-B, S6A-B

Please ensure that you provide the individual numerical values that underlie the summary data as they are essential for readers to assess your analysis and to reproduce it. The numerical data provided should include all replicates AND the way in which the plotted mean and errors were derived (it should not present only the mean/average values).

(D) Thank you for providing the raw spike data at the OSF repository (https://osf.io/vaet6/?view_only=2a3fb059695143178b323b6c6cbcb028). However, we note that the deposition is currently private, so we ask that you please make this data publicly available before publication.

(E) Please also ensure that each of the relevant figure legends in your manuscript include information on *WHERE THE UNDERLYING DATA CAN BE FOUND*, and ensure your supplemental data file/s has a legend.

(F) Please also provide a blurb which (if accepted) will be included in our weekly and monthly Electronic Table of Contents, sent out to readers of PLOS Biology, and may be used to promote your article in social media. The blurb should be about 30-40 words long and is subject to editorial changes. It should, without exaggeration, entice people to read your manuscript. It should not be redundant with the title and should not contain acronyms or abbreviations. For examples, view our author guidelines: https://journals.plos.org/plosbiology/s/revising-your-manuscript#loc-blurb

We expect to receive your revised manuscript within two weeks. 

*Published Peer Review History*

*Press*

Kind regards,

Richard

Richard Hodge, PhD

Associate Editor, PLOS Biology

rhodge@plos.org

Reviewer remarks:

Reviewer #1: Yang and Naya report on the activity of neurons in perirhinal cortex, area TE, parahippocampal cortex and hippocampus while macaque monkeys perform a visual associative memory task with both object and spatial components. They consider the task to tax object-location association memory. This is an interesting study that reports dissociations between perirhinal cortex (PRC) and hippocampus (HPC) encoding of task variables. Based on this information, Yang and Naya argue that the perirhinal cortex and hippocampus play different roles in semantic memory recall and/or retrieval.

The revised manuscript is much improved. The authors have responded adequately to my comments. Due to the substantial revision, I have a few additional comments that should be addressed. 

Specific comments:

Line 66: the PHC is usually considered to reside in the ventral pathway. In the Discussion (e.g., line 254), the authors explicitly refer to the PHC as being part of the MTL. The authors should consider rephrasing this - perhaps something like PHC is a ventral stream structure that is also well connected with the dorsal pathway? 

Lines 154-156: the authors need to back-off this statement regarding correlation with of the physiological findings with behavioral effects of lesions. There are many contrary findings in both humans and monkeys regarding participation of PRC in visual perception. Either remove the statement (starting with 'which is inconsistent') or acknowledge the contrary findings.

Lines 336-338: the sentence starting with 'Notably' is difficult to understand. Please recast or add explanation.

Lines 338-354: I gather that the authors are trying to address the referee 3's comment that the task (and therefore the physiological correlates) may tax episodic memory. Isn't there a simpler explanation? Clearly the monkeys have been trained for thousands of trials with a particular small set of objects and the one map. This argues against the idea that the learning and recall expressed here by the monkeys is episodic-like. If the authors (and editor) agree, I suggest they replace the new text with the simpler explanation. 

Line 371: it is not very helpful to refer to a manuscript 'in press' for supporting evidence. If the information in the cited ms. is providing essential evidence for this statement/conclusion, it should be explained. Alternatively, if the data in the present paper is sufficient then the citation could be dropped.

Reviewer #2: The authors adequately answered most of the reviewer's questions. However, not all were answered adequately. The following is an elaboration of the points that require further consideration.

The authors' answer to my previous Major Comment 3 is not convincing enough for me. In response to my previous comment, the authors wrote 'We apologize that the original manuscript might be misleading and that the reviewer might have misunderstood it as "the authors analyzed the 'first-person perspective effect of the retrieved location' during the choice-fixation period".' However, in the authors' revised manuscript, the phrase 'first-person perspective' appears six times in the main text and once in the abstract. Even a section titled 'First-person perspective of the retrieved location' is included in the Results chapter. Therefore, a proper analysis of the 'first-person perspective effect' is essential in this paper to ensure the validity of the data, analysis and interpretation of the study, whether it is based on 'choice-fixation period' data or 'item-cue period' data.

 I request the authors give a more detailed response to the previous Major Comment 3 by revising it again. In this paper, they must disclose results of the analysis of the "background-cue period" that I requested in my previous Major Comment 3 (probably in Supplementary Information). Disclosure of an analysis that includes all or part of the analytical methods I suggested in my previous Major Comment 3 would be preferable.

Minor

In response to my previous Minor Comment 2, the authors replied 'We are sorry we do not have independent evidence'. This point is important and, besides me, Reviewer #1 also expressed her/his concern on this point in her/his Comment 7. The authors have added a new discussion in this revision (Page 18, Line 331-Page 19, Line 339). However, it would be preferable to further discuss possible explanations for this phenomenon other than 'training history'.

Reviewer #3: The authors have addressed all my concerns. I would recommend to remove the word 'Conversely' from the abstract as the first and second sentences do refer to content comparable in nature.

---

## [Decision Letter · Decision Letter 3]

19 Apr 2023

Dear Dr Naya,

Thank you for your patience while we considered your revised manuscript "Sequential involvements of the perirhinal cortex and hippocampus in the recall of item-location associative memory in macaques" for publication as a Research Article at PLOS Biology. Please accept my sincere apologies for the long delays that you have experienced during this stage of the peer review process. This revised version of your manuscript has been evaluated by the PLOS Biology editors, the Academic Editor and by Reviewer #2. I have also provided some specific comments from the Academic Editor below the reviewer report (labelled as 'Comments from the Academic Editor'). 

The review is pasted below. As you will see, the reviewer still raises concerns that the analyses of the "first person perspective effect" and "context-cue period" have not been included in the manuscript, as well as noting that the Figures provided in the rebuttal should be included in the full manuscript. After discussions with the Academic Editor, we ask that you please clearly discuss the limitations of the work in terms of the statement of first-person, along the lines of Reviewer #2 (if performing or including this analysis is not possible).

We will then assess your revised manuscript and your response to the reviewers' comments with our Academic Editor aiming to avoid a further round of peer-review, although might need to consult with the reviewers again, depending on the nature of the revisions.

*Published Peer Review History*

*Press*

Kind regards,

Richard

Richard Hodge, PhD

Associate Editor, PLOS Biology

rhodge@plos.org

Reviewer #2: Comments on the R3 manuscript by Yang and Naya: 

1. The figure presented in "Response to Reviewer #2" (Figure X) is informative. This figure displays the results of a three-way ANOVA, including the PRC, HPC, and PHC, during the context-cue period. It is slightly different from the one I requested (i.e., I requested the data during "Context-cue plus Delay-2 period," because the "item-cue period" consists of "item-cue plus Delay-1 period" according to the authors' definition).

The figure (Figure X) should be shown in Supporting Information (a partial overlap with Figure S6a needs to be adjusted).

2. Figure X would be more informative if the results of the three-way ANOVA during the item-cue period were included in the figure in the same format as the results during the context-cue period (Expanded Figure X). Although some of the ANOVA results for "Item" are already shown in Figure 2b and Table S1, full disclosure of the ANOVA results including the "Context" and "Target" factors will help the reader understand the signal content of the PRC, HPC, and PHC neurons during each phase of this task.

I suggest including that figure (Expanded Figure X) in Supporting Information (the overlap with Figure 2b and Table S1 can remain unadjusted).

3. As I noted in my previous comments on the R2 manuscript, I believe that "proper analysis of 'first-person perspective effects' is essential in this paper to ensure the validity of the data, analysis, and interpretation of the study, whether it is based on 'choice-fixation period' data or 'item-cue period' data." With the results during the context-cue period shown in Figure X, I take the same position. I believe that an appropriate analysis of the context-cue period data is important, including all or some of the analytical methods that I had suggested in my Major Comment 3 on the R1 manuscript (Figure X shows that it is possible). It is favorable that the results of such analyses be disclosed in this paper, but if the authors plan to present those results in another paper, I look forward to reading it.

**COMMENTS FROM THE ACADEMIC EDITOR**

If the first-person analysis is not possible, they should give clear reasoning in the response why it is not possible (if this is part of a future work, it is ok). If they do not perform this analysis, then they should discuss in detail the limitation of the work in terms of the statement of first-person (in the discussion) along the line reviewer 2 raised. As for Figure X, they do not need to include this.

---

## [Editor Report · Decision Letter 4]

3 May 2023

Dear Dr Naya,

On behalf of my colleagues and the Academic Editor, Jozsef Csicsvari, I am pleased to say that we can accept your manuscript for publication, provided you address any remaining formatting and reporting issues. These will be detailed in an email you should receive within 2-3 business days from our colleagues in the journal operations team; no action is required from you until then. Please note that we will not be able to formally accept your manuscript and schedule it for publication until you have completed any requested changes.

PRESS

Kind regards, 

Richard

Richard Hodge, PhD

Associate Editor, PLOS Biology

rhodge@plos.org

PLOS
